

**A cobalt plume in the oxygen minimum zone of the Eastern Tropical South Pacific**
**N. J. Hawco,[1,2] D.C. Ohnemus,[3] J. A. Resing,[4] B. S. Twining[3] and M. A. Saito[2]**
[1] MIT/WHOI Joint Program in Oceanography/Applied Ocean Science and Engineering, Woods
Hole, MA, USA
[2] Department of Marine Chemistry and Geochemistry, Woods Hole Oceanographic Institution,
Woods Hole, MA, USA
[3] Bigelow Laboratory for Ocean Sciences, East Boothbay, ME, USA
[4] Joint Institute for the Study of the Atmosphere and the Ocean, University of Washington and
NOAA-PMEL, Seattle, WA, USA
*Correspondence to:* M. A. Saito (msaito@whoi.edu)



**Abstract.** Cobalt is a nutrient to phytoplankton, but knowledge about its biogeochemical cycling is limited, especially in the Pacific Ocean. Here, we report sections of dissolved cobalt and labile cobalt from the US GEOTRACES GP16 transect in the South Pacific. The cobalt distribution is closely tied to the extent and intensity of the oxygen minimum zone in the eastern South Pacific with highest concentrations measured at the oxycline near the Peru margin. Below 200 m, remineralization and circulation produce an inverse relationship between cobalt and dissolved oxygen that extends throughout the basin. Within the oxygen minimum zone, elevated concentrations of labile cobalt are generated by input from coastal sources and reduced scavenging at low $O_2$. As these high cobalt waters are upwelled and advected offshore, phytoplankton export returns cobalt to low-oxygen water masses underneath. West of the Peru upwelling region, dissolved cobalt is less than 10 pM in the euphotic zone and strongly bound by organic ligands. Because the cobalt nutricline within the South Pacific gyre is deeper than oligotrophic regions in the North and South Atlantic, cobalt involved in sustaining phytoplankton productivity in the gyre is heavily recycled and ultimately arrives from lateral transport of upwelled waters from the eastern margin. In contrast to large coastal inputs, hydrothermal vents along the Eastern Pacific Rise appear to be a minor source of cobalt. Overall, these results demonstrate that oxygen biogeochemistry exerts a strong influence on cobalt cycling.

**Keywords.** Cobalt, oxygen minimum zone, scavenging, GEOTRACES, hydrothermal vents, manganese oxides, phytoplankton, South Pacific, Peru Upwelling, micronutrient

## 1. Introduction

Cobalt is the least abundant inorganic nutrient in seawater and its scarcity may affect

phytoplankton growth in certain regions (Moore et al., 2013). In the high macronutrient waters of

the Costa Rica upwelling dome, for instance, Co and iron (Fe) amendments to surface seawater

increased phytoplankton production more than Fe alone, promoting growth of the

cyanobacterium *Synechococcus* (Ahlgren et al., 2014; Saito et al., 2005). While eukaryotic

phytoplankton mainly use cobalt to compensate for insufficient zinc (Sunda and Huntsman,

1995), populating the same enzymes with either metal (Yee and Morel, 1996), marine

cyanobacteria have an absolute growth requirement for Co that cannot be substituted and

suggests they may be more prone to limitation (Saito et al., 2002). Yet, the extent to which their





growth *in situ* is affected by cobalt scarcity ultimately depends on the processes that add Co to,
or remove it from, the surface ocean relative to other limiting nutrients.

Biological cycling of dissolved cobalt (dCo) is apparent in vertical profiles, showing uptake and
export in the surface and regeneration in the thermocline (Bown et al., 2011; Dulaquais et al.,
2014a; Noble et al., 2012). While dCo in the euphotic zone can be entirely bound by strong
organic ligands, a substantial portion of subsurface dCo is unbound and labile (10–50 %, Bown
et al., 2012; Ellwood and van den Berg, 2001; Saito and Moffett, 2001; Saito et al., 2005) and
therefore vulnerable to scavenging (Moffett and Ho, 1996). The similar ionic radii and redox
potentials of cobalt and manganese (Mn) cause dCo to be actively incorporated into bacterial
Mn-oxides, which sink from the water column and accumulate in marine sediments (Cowen and
Bruland, 1985; Moffett and Ho, 1996; Swanner et al., 2014). Below the euphotic zone, the
persistence of labile cobalt (LCo) throughout the Atlantic indicates that scavenging of dCo,
unlike Fe, is slow (Noble et al., 2012). On timescales of ocean circulation, however, scavenging
is responsible for decreasing dCo concentrations with depth and for the low ratio between dCo
and macronutrients in deep waters (Moore et al., 2013). As these deep waters are repackaged into
thermocline water masses and eventually brought to the surface (Sarmiento et al., 2011), the
upper ocean would become depleted in cobalt – as well as other hybrid metals like Fe – without
external sources that keep pace with scavenging (Bruland and Lohan, 2003; Noble et al., 2008).

Yet, the nature of marine cobalt sources is uncertain. In zonal sections of the North and South
Atlantic, sources appear to be concentrated along continental margins (Noble and Saito, in prep;
Noble et al., 2012). In the western Atlantic, dCo concentrations exceeding 100pM were



associated with the flow of Upper Labrador Seawater, likely gained through intense sediment
resuspension along the shelf or input prior to subduction (Noble and Saito, In prep). dCo in fresh
and estuarine waters can be 100–1000x greater than seawater (Gaillardet et al., 2003; Knauer et
al., 1982; Tovar-Sánchez et al., 2004) and terrigenous inputs from the American continent can be
clearly seen in lower salinity surface waters influenced by the Gulf Stream (Noble and Saito, in
prep; Saito and Moffett, 2002) and Amazon discharge (Dulaquais et al., 2014b). Yet, in both the
North and South Atlantic, a much larger dCo plume was associated with the oxygen minimum
zones along the Mauritanian and Namibian coasts (Noble and Saito, in prep, Noble et al., 2012).
Although these waters are not anoxic, the dCo plumes imply that $O_2$ over the continental shelf is
sufficiently low that reductive dissolution of Mn and Fe oxides in sediments releases a large flux
of dCo to the water column (Heggie and Lewis, 1984; Sundby et al., 1986). Drawing from large
inventories in the Atlantic OMZs, upwelling along eastern margins provides a large dCo flux to
the surface ocean. While surface dCo maxima from atmospheric deposition generally do not
appear in vertical profiles, this process may be important for regions that are isolated from
continental input or receive very high levels of dust (e.g. the Sargasso Sea, Dulaquais et al.,
2014a; Shelley et al., 2012).

To date, sectional datasets for dCo have been confined to the Atlantic and, as such, our
understanding of cobalt cycling may be biased by the dominant processes occurring there. In
comparison, the South Pacific receives considerably less dust deposition and river input
(Mahowald et al., 2005; Milliman and Farnsworth, 2011), but hosts a much larger and more
reducing oxygen minimum zone. Surface transects off Peru and the Costa Rica Dome suggest a
large source from upwelling (Ahlgren et al., 2014; Saito et al., 2004, 2005); however, profiles in



the tropical Pacific are sparse (Noble et al., 2008; Saito et al., 2014). We measured dissolved and
labile cobalt concentrations from over 750 samples collected onboard the 2013 US
GEOTRACES GP16 expedition across the South Pacific along 12° S, intersecting coastal
upwelling along the Peru margin, hydrothermal venting over the East Pacific rise, and
oligotrophic conditions near Polynesia (Fig. 1). Across this section, the distribution of dCo and
LCo follow the intensity of the oxygen minimum zone, with highest concentrations near the
South American shelf and low concentrations in both deep waters and oligotrophic surface
waters, matching OMZ-associated plumes observed in the Atlantic.

**2. Methods**
**2.1 Sample collection and handling**
Sampling on GP16 was conducted with a 24 position trace metal clean titanium rosette attached
to a non-metallic Kevlar cable designed for the U.S. GEOTRACES program (Cutter and
Bruland, 2012). An additional sample was collected from a surface towfish at each station.
Subsamples were collected in a Class-100 sampling van from 12 L Go-Flo bottles (General
Oceanics) and passed through 0.2 μM Acropack filters (Pall). All bottles were rinsed 3x with
sample seawater before being filled entirely, leaving no headspace. For samples analyzed at sea,
both dissolved and labile cobalt were analyzed from the same bottle. All samples were kept
refrigerated at 4° C until analysis in a HEPA filtered clean van. All of the LCo samples and more
than 90 % of dCo samples were analyzed at sea. Samples not analyzed at sea were preserved for
dCo immediately after sampling using metal-free gas adsorbing satchels (Mitsubishi Gas
Chemical, model RP-3K), using 3–4 satchels per 6 seawater samples. Gas-impermeable plastic



bags (Ampac) were heat sealed and were hand carried directly to Woods Hole at 4° C following
disembarkation.

**2.2 Cobalt determination by cathodic stripping voltammetry**
dCo and LCo were measured using a cathodic stripping voltammetry (CSV) method optimized
for organic speciation by Saito and Moffett, 2001. This method relies on the complexation of
inorganic Co species by a strong synthetic ligand, dimethylglyoxime (DMG, $K^{cond} = 10^{11.5 \pm 0.3}$),
which forms a bis-complex, $Co(DMG)_2$, with $Co^{2+}$ that readily absorbs to a hanging mercury
drop (Saito and Moffett, 2001). The $Co(DMG)_2$ complex is measured following a fast, $10V\ s^{-1}$
sweep that reduces both the Co (II) to Co(0) and the DMG to 2,3-bis(hydroxylamino)butane,
producing an $8 - 10$ electron decrease in current for each $Co(DMG)_2$ complex (Baxter et al.,
1998). The height of the $Co(DMG)_2$ reduction peak at -1.15 V is measured is directly
proportional to the Co concentration.

Triplicate scans of the seawater sample were followed by four standard cobalt additions (25 pM
per addition) and the slope of their linear regression (mean $R^2 = 0.998$) was used to calculate the
sample specific sensitivity (in $nA\ pM^{-1}$). The cobalt concentration was determined by dividing
the mean of the three baseline peaks by the sensitivity, and correcting for reagent volume. The
average deviation for these triplicate scans was 1.5 pM.

dCo analyses were conducted after a 1-hour UV oxidation procedure to remove strong organic
ligands that prevent Co binding by DMG. UV digestion was performed in 15 mL quartz glass
tubes using a Metrohm 705 UV digester (Metrohm USA). Temperature was maintained below



20° C to minimize evaporation losses. After UV digestion, 11 mL of sample was pipetted into
15mL polypropylene tubes and DMG and a buffering agent, EPPS, were added to final
concentrations of 400μM and 3.8mM, respectively. 8.5 mL of sample solution was added to a
Teflon analysis cup and mixed with 1.5 mL 1.5M $NaNO_2$, making a final analysis volume of 10
mL.

LCo was measured after >8 hour incubation of 11 mL of seawater with 400 μM DMG in a
Teflon cup. LCo is therefore the concentration that will readily exchange with DMG. After this
time, the sample was poured into an autosampler-compatible 15 mL poly-propylene tube
(separate from those used for dCo analyses) and EPPS was added to 3.8 mM.

**2.2.1 Preparing reagent and blanks**
All bottles and sample tubes were soaked for >1 week in the acidic detergent Citranox, rinsed
thoroughly with Milli-Q water (Millipore), filled with 10 % trace metal grade HCl (J.T. Baker) to
soak for 10 days, and rinsed thoroughly with ~10 mM TM-grade HCl. DMG (Sigma-Aldrich)
was purified by recrystallization in a 1 mM EDTA solution (Sigma-Aldrich). Crystals were
filtered, dried, and dissolved in HPLC grade methanol to a concentration of 0.1 M (Saito and
Moffett, 2001). EPPS (Fischer) and Sodium Nitrite (Millipore) were both dissolved in Milli-Q
water to 0.5 M and 1.5 M, respectively, and treated with separate batches of thoroughly cleaned
Chelex-100 beads (Bio-Rad) to remove background Co and Ni (Price et al., 1989). Standard
additions were generated by diluting a 1 ppm certified reference standard (SPEX Certiprep) with
10 mM HCl to a concentration of 5.00 nM. 50 μl of this solution was added to the 10mL sample
volume for each standard addition (25 pM addition).




To determine reagent blanks, Co-free seawater was generated by treating UV-seawater with
cleaned Chelex-100 beads. The seawater was then UV digested a second time to remove any
ligands leached during Chelex treatment. Any dCo measured in the Chelexed seawater derives
from addition of Co from analytical reagents. The mean blank for at sea analysis was
consistently low, $3.7 \pm 1.2$ pM (n=28). For analyses at Woods Hole, mean blank was $4.7 \pm 1.4$
pM (n=12). Blanks were subtracted from all measured values. Detection limits were calculated
as triple the standard deviation of the blank: 3.6 pM for at-sea analyses and 4.2 pM for samples
measured in Woods Hole.

**2.2.2 Automated cobalt analyses**
To accommodate a greater number of samples, our previous workflow (Noble et al., 2008) was
modified to incorporate fully automated sample analyses using the Metrohm 837 Sample
Processor autosampler. All measurements were performed using an Eco-Chemie µAutolabIII
system connected to a Metrohm 663 VA stand. A hanging drop mercury electrode (Metrohm)
was set to semi-hanging drop mode and accompanied by a 3M KCl/AgCl reference electrode and
glassy carbon auxiliary electrode. Scheduling and data acquisition were controlled using NOVA
1.8 software (Metrohm Autolab B.V). Automated delivery of seawater, sodium nitrite, and Co
standard to the analysis cup was accomplished by three dedicated Dosino 800 burettes
(Metrohm). Sample volume was increased to allow ~2 mL for conditioning tubing and analysis
cup prior of sample delivery.



Tubes containing 11 mL seawater, DMG, and EPPS were inverted several times and placed onto
a sampling rack where 8.5 mL of the mixture was dosed into the Teflon analysis cup. 1.5 mL 1.5
M sodium nitrite was added directly to the sample cup. Samples were purged with high-purity $N_2$
(>99.99 %) for 180 s and then conditioned for 90 s at -0.6 V. Scan sweeps were run at 10 V s$^{-1}$
from -0.6 V to -1.4 V. Before each analysis, the sample cup was rinsed fully with Milli-Q water
and 1 mL sample before measurement. Between uses, autosampler tubes, quartz vials, and
Teflon cups were rinsed with 10 mM HCl, Milli-Q water, and 1-2 mL of new sample. The
autosampler sample uptake line was rinsed with 10 mM HCl and Milli-Q when transitioning
from LCo analyses to dCo analyses.

We noticed a decrease in sensitivity of preserved samples relative to those analyzed at sea,
possibly caused by an increase in the sample pH during storage. Sensitivity was restored by
doubling the concentration of our buffering agent, EPPS, in the sample to a final concentration of
7.6 mM. We tested a broader range of EPPS additions in UV seawater and found the cobalt
concentration unchanged while the deviation between triplicate scans was reduced markedly by
the increase in sensitivity (data not shown). We tentatively attribute this decrease in sensitivity in
preserved samples to $CO_2$ adsorption by gas satchels, which would increase sample pH.

**2.2.3 Signal processing**
Analyses conducted at sea were characterized with a mild to moderate electrical interference that
mandated additional processing before peak height could be reliably measured (Fig. 2). We
opted for a simplified least squared fitting routine included in the NOVA software package that
conducts a 15-point weighted moving average – equivalent to a 36.9 mV analytical window –



according to a 2$^{nd}$ order polynomial. This method did not distort cobalt concentrations when
noise was low (Fig. 2a, b). A small fraction of scans (~3 %) were not adequately fit using this
routine and were instead smoothed using a 9-point linear moving average (22.1 mV window,
Fig. 2c), also included in NOVA. For all samples, peak height was measured manually to
minimize peak distortion due to added noise.

Subsequent analyses in the laboratory at Woods Hole were able to remove this signal by
increasing the current sampling step from 2.46 mV (341 points between -0.6V to -1.4V) to 4.88
mV (174 points) which eliminated the need for smoothing prior to sample analysis. We observed
good agreement between samples analyzed at sea and in lab, indicating that the smoothing
procedures applied at sea did not bias the data and that gas adsorbing satchels preserved original
concentrations (Noble and Saito, in prep).

**2.2.4 Intercalibration and internal laboratory standard**
All data reported in this manuscript have been submitted to the Biological and Chemical
Oceanography Data Management Center (BCO-DMO). Our laboratory continues to participate
in international intercalibration efforts through the GEOTRACES program in anticipation of the
release of the 2$^{nd}$ Intermediate Data Product, Summer 2017. The sampling scheme for GP16
included 2 overlapping samples per full depth profile where the shallowest sample of the deep
cast matched the deepest sample for the mid cast and the shallowest sample from the mid cast
matched the deepest sample from the shallow cast (i.e. a 36-point profile is composed of 34
discreet depths and 2 overlapping depths). Comparing overlapping samples collected at the same
depth and location on separate hydrocasts provides a measure of reproducibility. The average



difference between dCo analyses across 40 overlapping depth samples was 5.7 pM with a
median difference of 3.5 pM. For labile cobalt, average deviation was 2.1 pM (median of 2.0
pM, n=41). Least-squares regression of these samples yielded slopes close to 1(0.98 for dCo and
0.96 for LCo; y-intercept forced to 0), indicating good reproducability. Furthermore,
comparisons with other groups measuring dCo in the same samples reported here suggest strong
agreement between groups despite major methodological differences (Parker and Bruland,
personal communication).

Because acidified community reference materials such as the SAFe standards require a delicate
neutralization to pH 7.5–8 prior to analysis, a large batch of UV oligotrophic seawater was
generated prior to the cruise and used to access instrument performance during at sea analysis.
This consistency seawater standard was run ~3x per week, as were blanks, and values were
stable over several reagent batches for the duration of the cruise (4.5 ± 2.1 pM, n=28). SAFe
standard D1 was measured at sea (48.5 ± 2.4 pM, n=3) and fell within 1 SD of the consensus
value (46.6 ± 4.8 pM). SAFe standard D2 and GEOTRACES standard GSP were run at higher
frequency for analyses at Woods Hole. Our measurements of D2 (46.9 ± 3.0 pM, n=7) agreed
with consensus values (45.7 ± 2.9) and concentrations from our lab published previously (Noble
et al., 2012). While the GSP standard does not have a consensus value, our determinations (2.5 ±
2.0 pM, n=10) are within the range for SAFe S (4.9 ± 1.2 pM), which was collected at the same
offshore location as GSP. Acidified SAFe and GEOTRACES standards were neutralized with
concentrated ammonium hydroxide (Seastar), mixing the entire sample between drops, prior to
UV digestion. When base was added more quickly, measured Co was halved, presumably due to
adsorption or co-precipitation onto magnesium hydroxides formed during base addition. For



analysis of neutralized standards, we found a ~6:1 EPPS:NH$_4$OH (M:M) buffer improved pH
stability during analysis and removed significant baseline drift observed with samples solely
buffered with EPPS.

**2.3 Particulate metal analyses**
Particulate material collected from Go-Flo bottles was filtered onto acid-cleaned 0.45 μm
polyethersulfone filters (25 mm). Digestion protocol and analyses are identical to those used to
measure particulate metal concentrations during the North Atlantic GA03 cruise, described in
Twining et al. 2015. After filtration, filters were halved, digested at 135° C in sealed Teflon vials
containing 4 M HCL, HNO$_3$, and HF, dried, and redissolved in 0.32M HNO$_3$ before analysis.
pCo, pMn and pP concentrations were measured by ICP-MS (Element 2, Thermo Scientific),
calibrated using external multi-element standard curves, and corrected for instrument drift and
sample recovery by In and Cs internal standards. More detailed methods for this dataset can be
found elsewhere (Ohnemus et al., In Review).

**3. Results**
We report 680 determinations of dissolved cobalt (dCo) and 783 determinations of labile cobalt
(LCo) measured at sea, onboard the GP16 expedition in October–December 2013, as well as an
additional 140 measurements of dCo measured from preserved samples on land. Throughout the
GP16 transect, nutrient uptake and scavenging result in a hybrid-type profile for dCo (Fig. 3),
similar to dCo profiles from the Atlantic (Bown et al., 2011; Dulaquais et al., 2014b; Noble et
al., 2012; Noble and Saito, in prep) and North Pacific (Ahlgren et al., 2014; Knauer et al., 1982;
Saito et al., 2014). dCo ranged from <3 pM (below detection) in the South Pacific Gyre to 210



pM beneath the oxycline near the Peru Margin (Station 1). In the deep Pacific, concentrations
fell between 20–40 pM but increased slightly at deepest stations below 4500 m. These values are
much less than those observed in zonal transects surveying the North and South Atlantic (Noble
et al., 2012; Noble and Saito, in prep) but are similar to measurements in the Southern Ocean
(Bown et al., 2011), indicating that dCo is scavenged in the deep ocean along meridional
overturing circulation. Below 3000 m, dCo is somewhat lower east of the Eastern Pacific Rise
(EPR), and matches less oxygenated, older waters than in the western portion of the transect
(Fig. 4). While many profiles west of the EPR show considerable variation between 2000–3000
m suggestive of hydrothermal influence, the range is small (<10 pM) relative to background
concentrations (30–40 pM) and unlike the 50-fold excess of hydrothermal dFe and dMn above
background seawater measured at Station 18 (Resing et al., 2015).

dCo peaks in the mesopelagic, typically between 300–500 m. Towards the Peru shelf, this
maximum shoals and increases, following the position and intensity of the oxygen minimum
zone (OMZ, defined here as $O_2 < 20$ μM). Although the OMZ is several hundred meters thick
near the eastern margin (Fig. 3), dCo concentrations >100 pM are restricted to samples collected
just below the oxycline. Despite this narrow depth range, dCo >100 pM extends as far as 100°
W. For all depths below 200 m, dCo follows a negative linear relationship with $O_2$ (Fig. 5a).
Over the Peru shelf, maximum dCo was measured at the top of the OMZ and dCo decreased with
depth (except for the shallowest and most shoreward Station 2). Only at the western edge of the
section do dCo and $O_2$ decouple: the dCo maximum at station 36 is deeper (500–1000 m) than
the oxygen minimum (300–500 m), seemingly independent of the influence of the South Pacific
OMZ (Fig. 3).






All profiles show a surface or near-surface minimum that indicates biological uptake and export.
As a result, dCo is well traced by dissolved phosphate, $PO_4$, in the upper 200 m of the ETSP
(Fig. 5c). This relationship holds despite sharp transitions to high dCo in the oxycline near the
Peru shelf. Upwelling of $O_2$-depleted, $PO_4$-rich waters along the eastern boundary results in high
surface dCo, decreasing westward due to mixing and export. A secondary surface dCo maximum
marked a cyclonic eddy sampled at 89° W (Station 9, V. Sanial, personal communication), which
appeared to transport a shelf-like dCo and LCo signature for the upper 300 m into the offshore
OMZ (Fig. 6). Toward the South Pacific gyre, dCo in the euphotic zone falls below 10 pM.
While the lowest $PO_4$ was found in low salinity surface waters west of 140° W, minimum dCo
and deepest nutriclines corresponded to a southwestward excursion in the transect between
Stations 17–23 (109–120° W), which were accompanied by high salinities (>36) associated with
the eastern part of the subtropical gyre (Fig. 13). In contrast to the deep, smooth dCo nutricline
further to the east, stations at the western edge of the section (Stations 32, 34, and 36) contained
20 pM Co until ~50 m where concentrations decrease sharply surfaceward, resembling profiles
in the North Atlantic (Fig. 3; Noble and Saito, in prep).

The surface minimum in dCo is mirrored by a near-surface maximum in particulate cobalt (pCo)
from biological uptake throughout the GP16 section. The distribution of pCo (Fig. 6c) resembles
particulate phosphorus (pP), chlorophyll, and other indicators of phytoplankton biomass. Very
high pCo (>10 pM) was measured in the highly productive waters in the Peru upwelling
ecosystem while lower concentrations (2–4 pM) were found in oligotrophic surface waters. West
of 100° W, a secondary pCo maximum between 300–500 m overlaps with high particulate Mn



(pMn), reflecting Co incorporation into Mn-oxides in oxygenated thermocline waters. Elevated
pCo was also found at the top of the OMZ in the eastern half of the transect, corresponding with
high dCo from remineralization. High pP and low pMn in these samples suggest that pCo may be
present as biomass in anoxic bacterial and archaeal communities (Ohnemus et al., In Review),
rather than incorporation into bacterial Mn-oxides by co-oxidation. pMn increases sharply west
of 100° W, implying that pCo here is present as an authigenic phase (Fig. 9, Ohnemus et al., In
Review; Moffett and Ho, 1996).

dCo can be bound by extremely strong organic ligands that affect its reactivity (Ellwood and van
den Berg, 2001; Saito and Moffett, 2001). These ligands may be composed of degradation
products of the cobalt-bearing cofactor vitamin $B_{12}$ and may be stabilized following oxidation of
Co(II) to Co(III) (Baars and Croot, 2014). Unlike other metals such as Fe, dCo bound to natural
ligands is kinetically inert to ligand exchange (although some forms may still be bioavailable)
and strong Co(II) ligands are not in excess of dCo, largely due to binding competition with
nanomolar levels of labile nickel (Saito and Moffett, 2001; Saito et al., 2005). These properties
can result in a significant fraction of labile dissolved cobalt (LCo) that can be measured without
the UV-oxidation procedure necessary to measure dCo, especially in the mesopelagic (Noble et
al., 2012).

On GP16, the distribution of LCo is similar to that of dCo (Figs. 3, 4, 6). Except for samples
from the upper 50 m, dCo and LCo form a linear relationship ($R^2 = 0.88$) whose slope indicates
that ~33 % of dCo is labile (Fig. 7a). Major exceptions are confined to the highly productive
waters over the Peru shelf (Stations 1–6) where LCo is much lower than expected from dCo. In



these waters, LCo decreases in step with silicate (Fig. 7c). As in the North and South Atlantic
(Noble et al., 2012; Noble and Saito, in prep), LCo is undetectable in the surface ocean outside
of the waters influenced by upwelling (beyond 100° W, Fig. 6b). The absence of LCo from the
upper 300 m of the water column is deeper than corresponding gradients in the Atlantic,
suggesting cobalt depletion is more intense in the South Pacific.

In the deep Pacific (>3000 m), where dCo is low, LCo is undetectable. LCo remains low (<15
pM) in the mesopelagic, except where the OMZ is most intense (Fig. 4). Within the OMZ, LCo
maxima coincide with dCo maxima (Stations 1–15), but further to the west these LCo maxima
are much less intense and occur deeper than dCo maxima (Fig. 3). The LCo plume from the
OMZ extends deeper (below 2000 m) than the corresponding dCo (<2000 m), suggesting that
remineralization and scavenging affect these quantities in different ways. Slight secondary
maxima between 1500–2000 m (10–15 pM) appear in the center of the section on $\sigma_\theta = 27.7$–8 kg
m$^{-3}$ isopycnal layers (Fig. 4c, 105° W–115° W), perhaps tracing transport of LCo remineralized
in the eastern basin as these waters flow over the mid ocean ridge.

**4. Discussion**
**4.1 Basin-scale coupling between dCo and O$_2$**
The most striking aspect of the dCo distribution in the ETSP is the very high concentrations
present in the OMZ (Figs. 3–5). Similar distributions have been observed in both the North and
South Atlantic, where >100 pM dCo plumes corresponded to low oxygen waters underneath the
Benguela and Mauritanian upwelling systems (Noble et al., 2012; Noble and Saito, in prep). In
the North Pacific, profiles from the Costa Rica Dome (Ahlgren et al., 2014), the California



margin (Biller and Bruland, 2013; Knauer et al., 1982), and the Central Pacific along 155° W
(Saito et al., 2014) support an OMZ-cobalt plume there as well. Based on measurements from
these four OMZs, oxygen biogeochemistry seems to exert a major control on cobalt cycling
throughout the oceans.

Between the Atlantic and Pacific basins, the magnitude of the observed dCo plumes does not
appear to scale with minimum $O_2$. While offshore $O_2$ in the Atlantic OMZs exceeds 20 μM,
much of the ETSP is anoxic (Karstensen et al., 2008; Ulloa et al., 2012): Winkler titrations of
discreet samples measured on the GP16 cruise indicate minimum $O_2$ to be <5 μM, while *in situ*
sensors suggest true concentrations in the ETSP can be even lower (Thamdrup et al., 2012). Yet,
dCo in the ETSP occupies a similar 100–200 pM range reported for the North and South Atlantic
OMZs (Noble et al., 2012; Noble and Saito, in prep). Either the redox thresholds that affect
processes like water column scavenging or sedimentary release are met in the suboxic Atlantic as
well as the anoxic Pacific, or the apparent evenness in dCo concentration between OMZs results
from other factors besides $O_2$ (e.g. continental sources, dust, remineralization).

In the ETSP, tight coupling between dCo and $O_2$ results in the strong inverse relationship that
describes all samples below 200 m (Fig. 5a). In light of the nutrient-like dCo depletion in the
surface of the ETSP and elsewhere (Fig. 6, Ahlgren et al., 2014; Dulaquais et al., 2014b; Noble
et al., 2012), this negative correlation might be attributed to remineralization: dCo is returned to
the dissolved phase from a sinking biogenic phase following respiration (i.e. $O_2$ consumption).
The slope of the dCo:$O_2$ line (-0.33 μM $M^{-1}$, $R^2$ = 0.75 for 200–5500 m) might then represent the
biological stoichiometry of the exported organic material in the ETSP. However, Co:P ratios in



particulate material collected in the upper 50 m on GP16 indicate greater phytoplankton cobalt
utilization (median pCo:pP of 140 $\mu$M M$^{-1}$ ÷ 118 O$_2$:P M M$^{-1}$ = 1.2 Co:O$_2$ $\mu$M M$^{-1}$; Fig. 13C;
DeVries and Deutsch, 2014). If the dCo:O$_2$ trend is borne solely from remineralization, a greater
slope would be expected. The linearity in the dCo:O$_2$ relationship is also not reproduced upon
conversion of O$_2$ to apparent oxygen utilization (AOU = O$_{2,saturation}$ − O$_{2, measured}$, Fig. 5b, R$^2$ =
0.49), implying that other factors besides remineralization (such as circulation and scavenging)
shape the subsurface dCo distribution as well.

In the deep ocean, near-conservative mixing of dCo with high and low O$_2$ water masses probably
contributes to the observed dCo:O$_2$ relationship. The enormous depth range (>5000 m) described
by the linear dCo:O$_2$ relationship contrasts with the near-exponential decrease in
remineralization rates with depth (e.g. Karstensen et al., 2008). It is likely that deep Pacific
circulation acts to spread signals of local dCo remineralization throughout the water column,
aggregating a multitude of export stoichiometries and remineralization processes into a single,
coherent relationship across the basin. LCo is undetectable below ~2500 m and the shallower
slope of the LCo:O$_2$ trend (-0.11 $\mu$M M$^{-1}$, R$^2$ = 0.67) implies that the dCo:O$_2$ relationship is
driven mostly by strongly complexed species, which are less vulnerable to co-oxidation by Mn-
oxidizing bacteria in the water column (Moffett and Ho, 1996). Since the deep Pacific can be
broadly regarded as a mixture of oxygenated circumpolar waters and OMZs (especially from the
North Pacific), the linear dCo:O$_2$ relationship between 200 and 5500 m may reflect mixing of a
dCo pool that is largely inert to losses by scavenging.



In the upper 200 m, dCo is not well coupled with $O_2$ and almost all samples fall above the line
established by deeper samples (Fig. 5a). Near the South American margin, the dCo maximum in
the upper OMZ is more than double the 0 μM intercept of the dCo:$O_2$ relationship from deeper
waters (77 pM, Fig. 5a). Given their resemblance to profiles of excess $N_2$ from denitrification
(Chang et al., 2010), it is likely that both the dCo maximum and its decrease with depth are
driven by a combination of remineralization of sinking biogenic cobalt and lateral transport of a
coastal cobalt source (DeVries et al., 2012).

**4.2 Distinct surface and mesopelagic Co:P relationships**
In the upper ocean (0–200 m), dCo is linearly related to $PO_4$ (Figs. 5c and 7b), indicating that the
processes controlling $PO_4$ in the surface – upwelling, mixing, biological uptake and export – are
the main drivers of dCo as well. In the upper 50m, the dCo:$PO_4$ slope (69 μM:M, $R^2$ = 0.89, Fig.
7b) may describe export stoichiometry throughout the Eastern Pacific. That the surface dCo: $PO_4$
slope intercepts the highest dCo concentrations (below the 50 m depth range of the regression,
Fig. 5c) indicates that new cobalt sourced from the shelf is rapidly incorporated into biological
cycling and that the capacity for phytoplankton Co uptake is not overwhelmed by the order of
magnitude higher dCo in coastal waters relative to the open ocean. Culture experiments with
model diatoms and coccolithophores demonstrate this capacity (Shaked et al., 2006; Sunda and
Huntsman, 1995; Yee and Morel, 1996), deploying Co to zinc enzymes to maintain activity
when Zn becomes scarce. When Zn is limiting, Co quotas, as judged by open ocean
phytoplankton (Sunda and Huntsman, 1992; Twining and Baines, 2013), are 10–100x greater
than Co quotas when Zn is replete. Therefore, minor substitution of Zn quotas by Co (~10 %)



can double cellular Co levels in eukaryotes, resulting in nearly complete uptake of dCo from the
surface ocean.

A separate nutrient-like dCo:PO$_4$ trend arises from gradients of both elements in the open ocean
nutricline (Fig. 5c). The slope of the mesopelagic trend (16 µM:M, for 200–1000 m, Figure 5c) is
much less than that measured for the upper 50 m (69 µM:M). Due to considerable preformed PO$_4$
in deep waters, as well as elevated dCo:PO$_4$ ratios in the OMZ, the mesopelagic dCo:PO$_4$
regression is considerably less robust than in the surface ($R^2 = 0.21$), though the slope does
reflect dCo and PO$_4$ covariation in this depth range when PO$_4 < 2$ µM (Fig. 5c). Regardless, there
seems to be a fundamental mismatch between dCo:PO$_4$ from the upper water column (0–200 m)
and that observed deeper (200–1000 m). In the eastern margin, the surface and mesopelagic
dCo:PO$_4$ vectors are joined at 2.6 µM PO$_4$ by a near-vertical line that makes the dCo:PO$_4$ domain
triangular. Interpretation of this line depends largely on its perceived direction: a downward
vector can be a fingerprint of scavenging while an upward vector describes a cobalt source
(Noble et al., 2008; Saito et al., 2010). This ambiguity is clarified by examining dCo:PO$_4$
gradients within isopycnal surfaces, which strongly indicate a source at low O$_2$. In the ETSP, σ$_θ$
26.2 and 26.4 isopycnals host the upper OMZ and the oxygenated thermocline waters west of
100° W. Water masses on these surfaces can be distinguished on the basis of salinity; from the
GP16 dataset, mixing between salty and deoxygenated equatorial sub-surface waters (ESSW or
13° C water) with fresher, ventilated Sub-Antarctic waters is apparent (Fiedler and Talley, 2006;
Toggweiler et al., 1991). Oxygenated waters on σ$_θ$ = 26.2 and 26.4 show a tight coupling
between dCo and PO$_4$ (Fig. 8). For samples with < 20µM O$_2$, however, deviation from the oxic
dCo:PO$_4$ trend is always positive, indicating a dCo source within the OMZ. When oxygen is



low, dCo follows salinity. Mixing of high salinity (34.9–35.0), high dCo ESSW from the
northeast with low salinity, low dCo Subantarctic waters explains the dCo:salinity covariation on
these isopycnal surfaces. While ESSW is fed by the lower equatorial undercurrent (EUC), which
originates near Papua New Guinea and transports a large Fe and Al source eastward (Slemons et
al., 2010), it is low in dCo (as measured at 155° W, Hawco and Saito, unpublished; Saito et al.,
2014). When the EUC bifurcates near the Galapagos, it mixes with coastal waters north and
south of the equator (Fiedler and Talley, 2006; Stramma et al., 2010), where its high dCo
signature is likely acquired.

The isopycnal dCo:salinity relationship also implies cobalt scavenging in the OMZ is low. This
is not surprising given the thermodynamic barriers to $MnO_2$ formation at low $O_2$ (Johnson et al.,
1996; von Langen et al., 1997) and very low particulate Mn measured in the ETSP OMZ (Fig.
9). In the OMZ, both pCo:pP and pMn:pP ratios in the OMZ are consistent with micronutrient
use by microbial communities and resemble biomass collected in the euphotic zone on GP16
(Co:P = 0.5–4 x $10^{-4}$ M $M^{-1}$, Mn:P ~$10^{-3}$ M $M^{-1}$, Ohnemus et al., In Review). These low,
biomass-like pCo:pP and pMn:pP signatures in the ETSP OMZ are consistent with redox-
barriers to Mn oxidation at very low $O_2$ (Ohnemus et al., In Review).

Crossing the anoxic/oxic transition at 100° W in the thermocline ($\sigma_\theta$ 26.2–27.0, centered at 300
m) results in a factor of ten higher pMn concentrations and implies a redox threshold to Mn
oxidation in the mesopelagic (Fig. 9). Heterotrophic Mn-oxidizing bacteria are known to
incorporate Co by enzymatic co-oxidation into the Mn-oxide lattice and are prevalent throughout
the water column (Cowen and Bruland, 1985; Moffett and Ho, 1996). While particulate Co





profiles in the ETSP have a near-surface maximum from biological uptake (Fig. 7c), pCo
attenuates much less with depth than pP in oxic thermocline waters. Very high pCo:pP ratios (up
to $10^{-3}$ M M$^{-1}$) are found in the oxygenated thermocline but not in the OMZ ($\sigma_\theta$ 26.2–27.0, Fig.
9). The coincidence of high pCo:pP and high pMn throughout the in the mesopelagic is
consistent with pCo being present in an authigenic Mn-oxide phase, marking an important
transition between nutrient-like cobalt cycling in the surface ocean (where pCo is almost entirely
biogenic) to Mn-oxide driven scavenging at depth.

The stimulation of cobalt scavenging across the anoxic/oxic transition at 100° W was reflected in
a sharp decrease in LCo:PO$_4$ as scavenging removed LCo from the water column (Fig. 9).
Indeed, the same oxygenated thermocline samples with high pMn and pCo:pP are responsible for
the shallow vector in dCo:PO$_4$ space (16 μM M$^{-1}$, Fig. 5c). The offset between high surface and
low mesopelagic dCo:PO$_4$ is mirrored by the lower surface and higher mesopelagic pCo:pP.
While scavenging is often presumed to draw chiefly from metals in the dissolved phase, the
heterotrophic nature of Mn-oxidizing bacteria and their abundance in sediment traps hint that
Mn-oxidizing bacteria may access biogenic metal pools within sinking particles (Cowen and
Bruland, 1985). In such a case, pCo may be shunted directly from a biogenic to an authigenic
phase without being truly remineralized, preventing the equal return of dCo at depth relative to
that exported from the surface, as documented here by the disparity between deep and shallow
dCo:PO$_4$ slopes (Fig. 14). An important consequence of mesopelagic scavenging is that
ventilation of these waters by upwelling without an exogenous source (e.g. margin sediments)
would create conditions whereby dCo, relative to PO$_4$, is not supplied to the same extent it is
presently utilized and exported. Because these scavenged waters are relatively shallow and have



short ventilation ages (Fiedler and Talley, 2006), fluxes of cobalt to the South Pacific from
margin sources must be sufficiently rapid to balance scavenging losses.

**4.3 A major cobalt source from the Peruvian shelf**
The strong covariation between high dCo and low $O_2$ in the ETSP and the intersection of the
OMZ with the South American margin suggests that reducing sediments along the continental
shelf may be an important cobalt source. Sections from the North and South Atlantic (Noble et
al., 2012; Noble and Saito, in prep) and profiles from the North Pacific (Ahlgren et al., 2014;
Knauer et al., 1982) have resulted in similar assertions, but the coincidence of high
phytoplankton productivity along eastern margins also imprints signals from elevated
remineralization. This is certainly the case for the ETSP, where stations 2 and 3 on the Peru shelf
featured > 1.5 µg chlorophyll $L^{-1}$ in the euphotic layer, and > 4 µM nitrite throughout the OMZ
from intensified anoxic remineralization. As a result, all but one station along the shelf shows a
dCo maximum at the oxycline, rather than the benthic boundary layer (Fig. 10). The lone
exception, Station 2, is also the most shoreward, having respective dCo and LCo maxima of 159
pM and 59 pM at the deepest depth (110 m), indicating a flux of cobalt to the water column.

A survey of continental shelf sediments underlying the Peru OMZ found low Co/Al ratios (1.2 ±
0.3 x $10^{-4}$ g $g^{-1}$, Böning et al., 2004) relative to Andesitic and upper continental crusts (2.63 and
2.11 x $10^{-4}$ g $g^{-1}$ respectively, McLennan, 2001; Taylor and McLennan, 1995), requiring that
about half of the Co delivered to the continental shelf from crustal sources dissolved prior to
burial on the shelf. The only other element to have a similar depletion was Mn, which covaried
with Co across all samples in the Böning et al. study, consistent release of both metals by



reductive dissolution. Near-surface Co and Mn content were slightly higher in the shallowest
sediments (<150 m water column depth) and uniformly low at deeper locations (Böning et al.,
2004), implying that sedimentation outpaces dissolution of Co and Mn only in very shallow
water columns and/or proximal to input, which explains the lack of dissolved benthic maxima for
both elements beyond Station 2 (Fig. 10).

Positive correlations between dCo, LCo and dMn within the OMZ on the Peru shelf reflect a
shared source (Figs. 10, 11). The slope of the LCo:dMn relationship, $18 \pm 2$ mM M$^{-1}$ R$^2$ = 0.76,
is nearly identical to that in upper continental crust and Andesite (21–26 mM M$^{-1}$, McLennan,
2001; Taylor and McLennan, 1995), matching expectations that mineral dissolution should
provide should provide labile Co. However, the steeper slope for the dCo:dMn relationship ($42 \pm$
5 mM M$^{-1}$ R$^2$ = 0.67) exceeds crustal endmembers. Addition of a second, Co-enriched
component is needed to explain the observed relationship. Given the massive productivity over
the Peru shelf, biological export of dCo and dMn into the OMZ is a reasonable cause for the high
Co:Mn ratio in the shelf OMZ. From particulate material in the upper 40 m of shelf stations (1–
5), the Co:Mn ratio in biomass is 100–110 mM M$^{-1}$ (median and mean), ~5 times higher than
crust, and falling within the range reported for single cell analysis of phytoplankton cells (70–
400 mM M$^{-1}$, Twining and Baines, 2013). The combination, then, of a high biotic Co:Mn and a
lower ratio from a sedimentary source can produce the slope observed in the water column, but
requires remineralized dCo to be chiefly ligand-bound in order to preserve the near-crustal
LCo:Mn slope. The higher phytoplankton Co:Mn ratio relative to their shared sedimentary
source results in a nutrient trap that returns upwelled dCo to the OMZ more efficiently than dMn
and implies that input of dCo from the shelf is rapidly followed by biological utilization,



demonstrated in the transition from a dMn-like profile below the oxycline to a PO$_4$-like profile
above it (Fig. 10).

The Co/Al ratio in buried sediments on the continental shelf can provide a course measure of
how much Co has been released to the ocean. Sub-crustal Co/Al ratios in Peruvian sediments
between 9–14° S (Böning et al., 2004) match similar measurements in Chilean OMZ sediments
at 36° S (Co/Al = 1.3 ± 0.1 x 10$^{-4}$ g g$^{-1}$, Table 2, Böning et al., 2009) and the Gulf of California
(1.4 x 10$^{-4}$ g g$^{-1}$, Brumsack, 1989). The deficit between these values and continental crust (2.11 x
10$^{-4}$ g g$^{-1}$) implies that dissolution of crustal materials along the eastern margin provides a large
source of dCo and LCo to the Pacific. Ultimately, this source is needed to balance extremely
high Co/Al ratios in Pacific pelagic sediments, which collect Co scavenged from the water
column (e.g Dunlea et al., 2015; Goldberg and Arrhenius, 1958).

In contrast to depleted Co along the South American shelf, Co/Al in shelf sediments from the
western margin of the Pacific appears crustal (Table 2). Holocene records from the Pearl River
delta and shelf slope in the South China Sea ~20° N (Hu et al., 2012, 2013) show mean Co/Al of
2.2 and 2.1 x 10$^{-4}$ g g$^{-1}$, respectively, similar to sediments from the Gulf of Papua at 9° S (2.3 x
10$^{-4}$ g g$^{-1}$, Alongi et al., 1996). Crustal Co/Al in these sedimentary systems implies that most of
the Co provided from fluvial sediment delivery either does not dissolve or is quickly reburied by
water column Mn oxidation, rates of which can be very high in estuaries and coastal seas
(Moffett and Ho, 1996; Moffett, 1994; Sunda and Huntsman, 1987; Sunda and Huntsman, 1990).



It is likely that oxidizing conditions in the water column prevent reductive dissolution on the
western margin, leading to crustal Co/Al ratios in shelf sediments, while suboxic conditions on
the eastern margin mobilize Co, evident in depleted Co/Al ratios. Although sedimentary anoxia
releases Co bound in Mn oxides, even a thin layer of $O_2$ penetration into sediments results in a
near-zero diffusive flux into the water column (Heggie and Lewis, 1984). Bottom water
deoxygenation restores Co fluxes to the water column (Johnson et al., 1988, Sundby et al., 1986),
but Co sulfide minerals can also precipitate, analogous to 'Goldilocks' mechanisms for benthic
Fe release where flux is maximized when redox conditions are low enough to promote oxide
dissolution but still high enough to avoid pyrite burial (and by analogy, CoS; Scholz et al., 2014).
CoS burial is evident in high Co/Al content of Black Sea sediments (Brumsack, 2006) and
sulfide-rich pockets of Namibian sediments near Walvis Bay ($2.9 \pm 0.7 \times 10^{-4}$ g g$^{-1}$, Borchers et
al., 2005), despite more widespread Co/Al depletion in suboxic (but not sulfidic) terrigenous
sediments underneath the Benguela upwelling region (Bremner and Willis, 1993). Prevailing
suboxic conditions along the Namibian coast ultimately lead to extensive dCo, dMn, and dFe
plumes that reach across the South Atlantic basin (Noble et al., 2012). Similarly, depleted
sedimentary Co/Al on the Peruvian margin and high dCo in the water column perhaps reflect
sustained anoxia that, in the present, is unlikely outside the domain of OMZs.

Can a terrigenous cobalt source account for the observed OMZ plume? Because lithogenic
sediments along the Peru margin are delivered primarily by rivers (Scheidegger and Krissek,
1982), we can estimate a dCo flux to OMZ waters as the product of the fluvial sediment delivery
to the continental shelf and the difference in Co/Al ratios between original rocks and buried shelf
sediments:



$$\text{Co flux}_{\text{suboxic}} = \left( \frac{Co}{Al}_{\text{crust}} - \frac{Co}{Al}_{\text{suboxic sediments}} \right) * \% \, Al * F_{\text{fluvial}} \tag{1}$$
where $F_{\text{fluvial}}$ is the riverine flux of terrigenous sediments from Ecuador, Peru and Northern Chile
to oxygen-depleted coastlines in the ETSP. If this supply is approximately 200 MT year$^{-1}$ (Lyle,
1981; Milliman and Farnsworth, 2011), the Co deficit in Peruvian sediments from Böning et al.,
2004 corresponds to a 2.5–4.6 x 10$^7$ mol per year flux from the South American shelf, depending
on the crustal endmember applied. When scaled to the size of the ETSP OMZ (2.2 x 10$^6$ km$^3$
defined at 20µM, Fuenzalida et al., 2009) a terrigenous cobalt supply of 11–21 pM year$^{-1}$ would
be expected.

The extent to which the coastal flux and dCo inventory are in agreement depends on the
residence time of OMZ waters. Models and CFC distributions from WOCE imply an
approximately decadal recirculation time in OMZ waters relative to mesopelagic gyre circulation
in the ETSP (Deutsch et al., 2001, 2011). Integrating our terrigenous Co flux estimate over 10
years yields an expected concentration of 120–230 pM within the OMZ. This is of similar
magnitude, but greater than the concentrations measured on the GP16 transect (mean of $100 \pm 30$
pM). The difference between estimated and actual dCo inventories in the OMZ is probably due
to upwelling and advection by surface currents, readily seen in the dCo section (Fig. 6), which
carries the remainder to the gyre.

We can compare the calculated sedimentary flux to an expected flux from aerosol dust
dissolution. Aeolian deposition is extremely low over the South Pacific basin (Mahowald et al.,
2005), except immediately offshore of Peru, where dust from the Altiplano interacts with the
prevailing northward winds (Prospero and Bonatti, 1969; Scheidegger and Krissek, 1982). Model



results (Mahowald et al., 2005) suggest that deposition does not exceed 0.5 g m$^{-2}$ yr$^{-1}$, except
very close to the coastline. Using this estimate, crustal cobalt abundances, and the aerial extent of
the OMZ ($9.8 \times 10^6$ km$^2$, Fuenzalida et al., 2009), we can estimate an aerial flux of Co from dust
to be $1.4 \times 10^6$ mol Co per year. A 10% fractional solubility for Co (Shelley et al., 2012)
indicates a soluble cobalt flux from dust of 0.065 pM year$^{-1}$, ~0.5 % of the expected sedimentary
flux. Over a decade, dust deposition accounts < 1 pM of the OMZ dCo plume. dCo profiles also
lack surface maxima near shore despite corresponding features for dissolved Al and Mn at
Stations 1 and 5 (Fig. 10, Resing et al., 2015). Fluvial sediment delivery to the margin, therefore,
is a much more plausible source for the elevated dCo in the ETSP OMZ.

**4.4 An inefficient cobalt source in hydrothermal vents**
Hydrothermal venting along the Eastern Pacific Rise (EPR) provides a major source of dFe and
dMn to the deep South Pacific (Resing et al., 2015) where nanomolar concentrations of both
metals were measured between 2000–3000 m at the ridge crest and concentrations exceeded
background values for several thousand kilometers westward. In contrast, dCo concentrations are
only slightly elevated at the ridge crest (Station 18, Fig. 12), reaching 36 pM at 2400m (against a
background of ~25 pM); at the same station, dFe and dMn both exceeded 15 nM (background <1
nM). Unambiguous hydrothermal input is evident from the LCo profile, which peaks at 14 pM at
the dCo maximum (Fig. 12), roughly consistent with a 10,000-fold dilution of high temperature
endmember sources containing 100–1000 nM dCo (Lupton et al., 1985; Metz and Trefry, 2000).

However, both dCo and LCo maxima are offset from dFe and dMn plumes. At station 18, dFe
and dMn peak at 2500–2600 m. At this depth, LCo is undetectable and dCo values are at – or




slightly lower than – background levels, suggesting that Mn and/or Fe scavenging in the heart of
the hydrothermal plume has removed most of the hydrothermal Co from the water column before
being transported away from the ridge crest (Fig. 12).  Indeed, Co is strongly associated with Mn
phases in near-axis metalliferous sediment in the EPR at 14° S (Dunk and Mills, 2006). The
position of the LCo maximum above the dMn and dFe maxima probably reflects lower
scavenging rates outside the main plume, which may spare a fraction of the hydrothermal Co
source from an otherwise immediate and total removal. Even without scavenging losses, global
hydrothermal Co fluxes (2.2 Mmol year$^{-1}$; Swanner et al., 2014) are 10–25x less than our
estimated source from the Peru Shelf alone, highlighting the importance of upper ocean sources
in maintaining the dCo inventory.

**4.5 Cobalt scarcity in the euphotic zone**
The combination of eastern boundary upwelling and a continental source produces large dCo
gradients across the surface of the South Pacific Ocean (Fig. 13). Westward, decreasing surface
dCo results from phytoplankton uptake and export, reflected in strong correlations with $PO_4$, as
well as mixing with low dCo waters from the subtropical gyre. It is interesting to note that the
intercept for the dCo:$PO_4$ relationship is negative (-12.8 ± 2.6 μM M$^{-1}$, Fig. 7b); cobalt was
depleted before $PO_4$. This is opposite to what was observed in the Sargasso Sea, where extreme
$PO_4$ scarcity results in a positive dCo:$PO_4$ intercept (Jakuba et al., 2008). The dCo nutricline in
the South Pacific gyre (~200 m, Figs. 3, 6) is also deeper than corresponding features in the
North and South Atlantic (Noble et al., 2012; Noble and Saito, in prep). Because winter mixed
layers in the tropical South Pacific do not exceed 100 m and strong haloclines separate the
oxygen minimum layer from the surface (Fiedler and Talley, 2006), convective overturn does not



reach the dCo nutricine at 150–250 m. Low vertical cobalt supply makes the South Pacific gyre
an interesting counterpoint to the Sargasso Sea, which experiences deep winter mixed layers and
higher dCo (10–30 pM, Dulaquais et al., 2014b; Jakuba et al., 2008; Noble and Saito, in prep),
and emphasizes the importance of lateral supply mechanisms, especially eastern boundary
upwelling, in maintaining the surface dCo inventory (Fig. 6, Saito et al., 2004).

While the South Pacific is thought to be limited by iron and nitrogen (Moore et al., 2013; Saito et
al., 2014), the extremely low dCo measured here implies that it may be important as well.
Because marine cyanobacteria such as *Prochlorococcus* and *Synechococcus* have an absolute Co
requirement (Saito et al., 2002; Sunda and Huntsman, 1995), they are vulnerable to limitation.
Indeed, a *Synechococcus* bloom in the Costa Rica Dome was found to be co-limited by both iron
and cobalt (Saito et al., 2005). Despite dCo concentrations below 10 pM (sometimes below the
3pM detection limit), the South Pacific gyre contains a significant *Prochlorococcus* population –
evident in the high proportion of divinyl chlorophyll A to total chlorophyll (Fig. 13b). As LCo
was undetectable beyond 100° W, biological uptake must occur either by accessing strongly
bound dCo or through fast cycling of LCo at very low steady-state concentrations. On GP16,
particulate Co concentrations in the upper 50 m were steady ($3.5 \pm 1.2$ pM) and sometimes
equaled dissolved concentration. Compared to low pCo:dCo ratios observed in the South
Atlantic (<1:12, Noble et al., 2012), the high ratio in the South Pacific gyre (~1:3, Fig. 13a)
indicates that resident *Prochlorococcus* are extremely well-adapted to widespread dCo scarcity.

Unlike its near uniform relationship with dCo in the underlying OMZ, LCo measured on GP16 is
low relative to dCo in the surface ocean (0–50 m), especially along the Peru Margin (Fig. 6b,



13c). This might result either from microbial production of cobalt ligands – as observed in a
*Synechococcus*-dominated community in the Costa Rica Dome (Saito et al., 2005) – or if LCo is
the preferred species for uptake. The latter has been demonstrated in culture with model
eukaryotic algae (Sunda and Huntsman, 1995), where free Co ion is acquired by high affinity
Zn-transporters. Because the Peru upwelling region is dominated by diatoms (Bruland et al.,
2005), preferential uptake of free Co ion by these organisms is realistic and is corroborated by a
strong correlation between LCo and dissolved silicate (Si) in the upper 50 m of the section ($R^2$ =
0.90). As a result of diatom-driven export, LCo in shelf surface waters is nearly depleted (1–12
pM) despite high concentrations of dCo (40–80 pM). Very low surface LCo at 12° S on GP16
contrasts with previous observations showing high concentrations of LCo in the Peru upwelling
region during August–September 2000 (Saito et al., 2004). Between 5–10° S, much higher
surface dCo was measured in freshly upwelled waters (up to 315 pM) and dCo was >50 % labile
in surface transects (Saito et al., 2004). Surface dFe during August–September 2000 was also
higher between 5–10° S than on GP16 at 12° S (Bruland et al., 2005; Resing et al., 2015).
Decreasing surface dFe from North to South followed decreasing gradients in shelf width and
fluvial sediment supply (Bruland et al., 2005; Milliman and Farnsworth, 2011), implying that the
high dFe was due to stronger benthic sources to the North. Because coastal sources are expected
to provide labile cobalt (e.g. Fig 11), the high concentrations of LCo measured between 5–10° S
during August–September 2000 indicate a similar gradient in coastal dCo input and LCo
availability in surface waters.

Spatial and temporal variability of margin dCo sources may ultimately affect carbon flow
through the Peru upwelling ecosystem. Considering the very low dissolved Zn in surface waters



on GP16 (<100 pM east of 90° W, S. John personal communication) and that >95 % of the dZn
is typically complexed by organic ligands (Bruland, 1989), coastal diatoms off of Peru may be
subject to diffusion limitation when free Zn ion falls below a 1–10 pM threshold (Sunda and
Huntsman, 1992). Because Co can replace Zn in carbonic anhydrase enzymes  (Sunda and
Huntsman, 1995; Yee and Morel, 1996), LCo supplied from the margin may maintain fast rates
of carbon fixation and export in the Peru Upwelling region despite low dZn. The relatively low
concentrations measured on GP16, however, imply that the LCo supply may not always be
sufficient.

**5. Conclusions**
The basin scale association between high dCo and low $O_2$ throughout the GP16 section testifies
to the importance of redox chemistry and remineralization in maintaining the dCo distribution in
the Eastern South Pacific (Fig. 14). High dCo and LCo on the Peru shelf match depleted Co
content reported in Peru shelf sediments and indicate a large source to the water column (Böning
et al., 2004). Correlations between dMn and LCo in anoxic shelf waters, and crust-like Co/Al
ratios in oxic western boundary sediments suggest that margin cobalt sources are redox sensitive
and that the sustained presence of the OMZ on the Peru margin amplifies coastal Co fluxes.
Additionally, low oxygen in offshore OMZ waters suppresses particulate Mn accumulation and
Co scavenging, evident in isopycnal dCo-salinity relationships. When mixing introduces
sufficient $O_2$ to stabilize Mn oxides, high particulate Co:P ratios in the mesopelagic accompany
10-fold higher particulate Mn. Scavenging in mesopelagic waters limits the full return of Co to
the dissolved phase during remineralization, resulting in low dissolved Co:P ratios relative to that
exported from the euphotic zone (Fig. 14). Oxidative scavenging also seems to limit the flux of



Co from hydrothermal venting over the East Pacific rise, further emphasizing the Peru margin as
the most important Co source to the South Pacific.

The high dCo within the OMZ also leads to a large flux to the surface ocean by upwelling along
the Peru margin, where it is readily accessed by phytoplankton. Strong correlations with
phosphate across the surface ocean emphasize its significance as a micronutrient. Preferential
uptake of LCo over the Peru shelf by diatoms generates a linear relationship with silicate and Co
may be critical to sustaining carbonic anhydrase activity and $CO_2$ fixation during blooms. Given
that deep nutriclines in the South Pacific Gyre limit dCo supply from vertical mixing outside of
the upwelling zone, phytoplankton Co nutrition depends largely on lateral supply from the
Eastern Margin.

Ultimately, the dCo inventory in the South Pacific – and its availability to surface phytoplankton
– may be changing considerably as the size and intensity of the OMZ fluctuate. Recent warming
and stratification appear to have expanded the volume of low oxygen waters in the tropics
(Stramma et al., 2008, 2010). As such, dCo inventories may increase as lower $O_2$ hinders
mesopelagic Mn oxide production and scavenging. However, decreased wind-driven upwelling
and carbon export may cause anoxic waters along the shelf to contract (Deutsch et al., 2011,
2014) and decrease the Co flux from coastal sediments. Changes in the dCo inventory then
depend on the relative redox sensitivities of the margin Co source versus offshore scavenging in
the mesopelagic, neither of which are well understood. Considering the 100–200 year residence
time of Co in the ocean (Saito and Moffett, 2002), feedbacks on the surface dCo supply may
manifest more quickly than for other nutrients and may alleviate or exacerbate any existing Co



limitation. Improved definition of biological Co limitation thresholds and efforts to reconstruct
the Co cycle in past climates may resolve whether future changes in OMZ structure will have
meaningful impacts on phytoplankton nutrition in the coming century.

**Author Contributions**
NJH, DCO and JAR participated on the EPZT cruise. NJH measured dCo and LCo. JAR
measured dMn; DCO and BST measured particulate Co, Mn, and P. NJH and MAS prepared the
manuscript with contributions from all authors.

**Acknowledgements**
We thank the Captain and Crew of the RV *Thomas G. Thompson* and the entirety of the science
party aboard the GP16 cruise, especially chief scientists Jim Moffett and Chris German. We also
thank Claire Parker and Cheryl Zurbrick for enormous efforts in sample collection, Greg Cutter
and Geo Smith for operating the trace metal rosette and Tow-Fish, respectively, and Sara
Rauschenberg and Rob Sherrell for particle sampling. Susan Becker, Melissa Miller, and Rob
Palomares contributed nutrient, oxygen, and salinity data. The technical and logistical expertise
of Dawn Moran and Matt McIlvin is unparalleled. We appreciate the efforts of the
GEOTRACES office in coordinating the GP16 expedition and thank Luca Pini and Mike
Kubicsko of Metrohm Autolab for assistance with voltammetry. This is PMEL publication
#4475 and JISAO publication 2648. This work was funded by NSF awards OCE-1233733 to
MAS, OCE-1232814 to BST, and OCE-1237011 to JAR.




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

**Table 1.** Blanks and standards used during analyses.

|  |  | Co, pM | +/- | n | Consensus |
|---|---|---|---|---|---|
| At sea | Blank | 3.7 | 1.2 | 28 | |
| Oct – Dec, | Lab SW | 4.5 | 2.1 | 28 | |
| 2013 | D1 | 48.5 | 2.4 | 3 | 46.6 +/- 4.8 |
|  |  |  |  |  | |
| At WHOI, | Blank | 4.7 | 1.4 | 12 | |
| Sept – Nov, | GSP | 2.5 | 2.0 | 10 | 4.9 +/- 1.2** |
| 2014 | D2 | 45.0 | 2.7 | 7 | 46.9 +/- 3.0 |
|  | GSC | 77.7 | 2.4 | 4 | *** |

*Collected from South Pacific surface seawater Nov. 2011
**Refers to SAFe standard S, which was collected at the same location
***No consensus


**Table 2.** Co/Al ratios in sediments from different redox regimes.

| Location | Co / Al x 10⁻⁴ (g g⁻¹) | | Reference |
|---|---|---|---|
| **Crust** | | | |
| Upper continental crust | 2.11 | | McLennan 2001 |
| Andesitic crust | 2.63 | | Taylor and McLennan 1995 |
| | | | |
| **Eastern Boundary sediments** | | | |
| Peru Upwelling Sediments | 1.2 ± 0.3 | 9–14° S | Boning et al. 2004 |
| Chile upwelling Sediments | 1.3 ± 0.1 | 36° S | Boning et al. 2009 |
| Gulf of California | 1.4 | | Brumsack 1989 |
| Namibian Shelf | 1.0 ± 0.3 | 17–25° S, 'Terrigenous' | Bremner and Willis 1993 |
| **Sulfidic Sediments** | | | |
| Namibian diatom belt | 2.9 ± 0.7 | Near Walvis Bay, 22.5° S | Borchers et al. 2005 |
| Black Sea | 3.8–6.2 | | Brumsack 2006 |
| **Western Boundary Sediments** | | | |
| Papua New Guinea | 2.3 | 8° S, Gulf of Papua | Alongi et al. 1996 |
| Pearl River Delta | 2.2 ± 0.4 | 22° N, (10–0 ka) | Hu et al. 2013 |
| South China Sea Shelf Slope | 2.1 ± 0.2 | 20° N, 2037m (14–0 ka) | Hu et al. 2012 |
| **Deep Ocean Sediments** | | | |
| South Pacific Gyre | 35 (2.9–101) | 22° S–32° S, 100–0 Ma | Dunlea et al. 2015 |
| Pelagic Pacific | 17 (2–58) | 50° N–20° S | Goldberg and Arrhenius 1958 |





**Figure Captions**
**Figure 1.** The GP16 transect in the tropical South Pacific. Red circles indicate sampling stations.
Dissolved oxygen from WOCE is plotted in blue and 10 μM contours are shown between 0–60
μM $O_2$. Station number increases sequentially westward, with the exception of Station 1.

**Figure 2**. Signal processing of voltammetry scans. Varying instrumental noise imprinted
negative current excursions during measurement and necessitated data smoothing to correctly
integrate the $Co(DMG)_2$ reduction peak at -1.15 V. For mild (a) and moderate (b) noise levels, a
$2^{nd}$ order, 17-point smoothing was applied (red line, 97 % of scans). Increases in noise caused
this routine to overestimate peak height (c) and a first order, 13-point smoothing was applied
instead (~3 % of scans, blue line).

**Figure 3**. Profiles of dissolved cobalt (dCo, closed circles), labile cobalt (LCo, open circles) and
$O_2$ (grey lines) across the South Pacific. Upper panels show a 0–1000 m depth range; bottom
panel show full profiles. dCo and LCo are highest close to the Peru Margin (Station 1) and
decrease westward. $O_2$ follows the opposite trend. The small peak at 2400 m at Station 18 shows
peak hydrothermal input from the East Pacific Rise. Note that the dCo and LCo scales in the
upper panel are adjusted to highlight gradients and differ from the lower panels.

**Figure 4.** Dissolved oxygen (a), dissolved cobalt (b) and labile cobalt (c) sections along GP16,
projected on a longitudinal axis. Note high dCo and LCo stem from the Peru margin and overlap
with the low $O_2$. Interpolations were made using Ocean Data View with DIVA gridding, with



negative gridded values suppressed. The signal to noise ratio was set to 15 for dissolved and
labile cobalt. Signal to noise for $O_2$ was set to the default, 50.

**Figure 5.** Coupling between dissolved cobalt with $O_2$ (a), AOU (b), and $PO_4$ (c). Below 200 m
(red circles), dCo shows a decreasing linear trend with dissolved oxygen that is obscured upon
conversion of $O_2$ to apparent oxygen utilization, AOU, and a weak relationship with $PO_4$. In the
upper 200 m (blue circles), dCo and $PO_4$ are strongly coupled but dCo shows no relationship
with $O_2$ or AOU. Trend lines in (c) show best fit for 0–50 m in particulate Co:P (146 μM $M^{-1}$,
dotted line), 0–50 m dissolved Co:P (69 μM $M^{-1}$, blue line, see Fig. 7b) and 200–1000 m
regression for dissolved Co:P (16 μM $M^{-1}$, red line). Processes affecting these plots are described
in vector legends.

**Figure 6.** Dissolved cobalt (a), labile cobalt (b), and particulate cobalt (c) gradients in the upper
500 m of the ETSP. White lines in both panels show dissolved $PO_4$ contours at 0.5 μM
increments. Interpolation was conducted using weight averaged gridding with 40 and 46 ‰
length scales in the x and y direction, respectively.

**Figure 7**. (a) The relationship between dissolved cobalt (dCo) and labile cobalt (LCo) in the
South Pacific. LCo increases linearly with dCo with a slope of 0.33 (black dots), except for the
upper 50 m (red dots), where samples fall below this trend due to preferential depletion of LCo
by phytoplankton. (b) In the 0–50 m range, dCo strongly correlates with phosphate ($R^2 = 0.89$).
(c) LCo is preferentially removed from surface waters (0–50 m) and tracks silicate ($R^2 = 0.90$).





**Figure 8**. Transition in dCo cycling at the OMZ boundary in the upper South Pacific
thermocline. (a) Isopycnal windows centered at $\sigma_\theta = 26.2 \pm 0.1$ kg m$^{-3}$ (circles) and $\sigma_\theta = 26.4 \pm$
0.1 kg m$^{-3}$ (triangles) show PO$_4$-coupled cycling in oxygenated waters (blue) but not in OMZ
waters with O$_2$<20 µM (red). (b) In the OMZ, dCo follows salinity, indicating mixing between a
high dCo endmember at a salinity of 35.0 and a fresher water mass that is low in dCo.

**Figure 9**. Redox control of Co and Mn scavenging. Within mesopelagic waters ($\sigma_\theta = 26.2$–27.0
kg m$^{-3}$, mean depth of 300 m), high O$_2$ in ventilating water masses result in a sharp redox
boundary at the edge of the OMZ (red circles, scale in µM). Particulate Mn (pMn) increases
across the oxic/anoxic boundary at 100° W (blue circles, in nM) and imply stabilization of Mn
oxides. Both pMn:pP and pCo:pP (cyan and pink circles, respectively; units are M M$^{-1}$) increase
across the OMZ boundary, exceeding predicted values from remineralization of biogenic
material from the surface ocean ( 1.26 x 10$^{-3}$ mean pMn:pP and 1.57 x 10$^{-4}$ pCo:pP for 0–50 m
(M M$^{-1}$), cyan and pink lines, respectively). Dissolved phase dCo:dPO$_4$ (black circles) and
LCo:PO$_4$ (white circles) also decrease west of the OMZ boundary show scavenging of the dCo
and LCo in the mesopelagic (units are M M$^{-1}$).

**Figure 10.** Profiles of dissolved cobalt (dCo, black), PO$_4$ (red) and dMn (blue) over the Peru
shelf at 12° S. The oxycline (grey bar, defined by the first sample where O$_2$<10 µM) marks the
transition between the OMZ and oxygenated surface waters. Station 2 is the closest to the coast.
Note the similarity between dCo and PO$_4$ above the oxycline and the transition to a dMn-like
profile beneath it.



**Figure 11.** Cobalt and Mn in the Peru shelf OMZ (GP16 Stations 1–5, $O_2$<20 uM). Dissolved

cobalt (dCo, closed circled) and labile cobalt (LCo, white) follow positive linear relationships

with dMn. The LCo slope (18 mM $M^{-1}$) approximates the Co:Mn ratio in upper continental crust

and Andesite (red and pink lines 21 and 26 mM $M^{-1}$, respectively), suggesting it derives from a

shelf source. The mean pCo:pMn ratio from phytoplankton dominated particles collected in the

upper 40 m over the shelf (blue line, 105 mM $M^{-1}$) is greater than dCo:dMn slope (42 mM $M^{-1}$),

indicating that dCo and dMn concentrations in the Peru shelf OMZ represent a combination of

biomass remineralization and sedimentary input.

**Figure 12.** (a) Dissolved cobalt (dCo, black circles) and labile cobalt (LCo, white) in the East

Pacific Rise hydrothermal plume. Profiles are from Station 18 at the EPR ridge crest at 113 °W.

(b) dMn (blue lines) and dFe profiles (red) replotted from Resing et al. 2015 clearly show

hydrothermal input. Grey shading below 2250 m indicates area of hydrothermal influence where

dFe and dMn are >1 nM.

**Figure 13.** (a) The particulate to dissolved ratio of cobalt (pCo:dCo, in M $M^{-1}$, pink circles) in

near-surface samples (0–50m) measured on GP16; a value of 1 indicates equal concentrations in

each phase. (b) The ratio of divinyl chlorophyll A to total chlorophyll (green circles), a proxy for

*Prochlorococcus* abundance. (c) The near-surface distribution of dissolved cobalt (dCo, black

circles) and labile cobalt (LCo, white circles), showing higher concentrations near the Peru

margin (<80° W) and very low dCo to the west.



**Figure 14.** A schematic cross-section of the cobalt cycle in the Eastern Tropical South Pacific.
Black arrows describe idealized physical circulation, showing upwelling near the Peru margin,
advection westward and subduction in the South Pacific gyre. Biological Co export is shown in
the red-hashed arrows, and solid and dashed red arrows show remineralization and scavenging
respectively. The margin source is shown as a red-outlined arrow. These vectors are also plotted
on idealized oxygen and phosphate axes, using the same color scheme, to show how these
processes appear in $Co:O_2$ and $Co:PO_4$ space.



# Figure 1

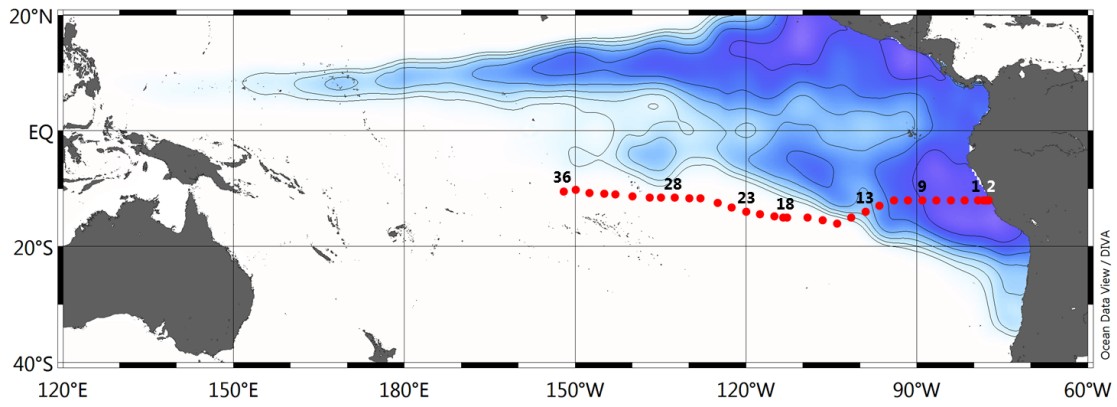





# Figure 2

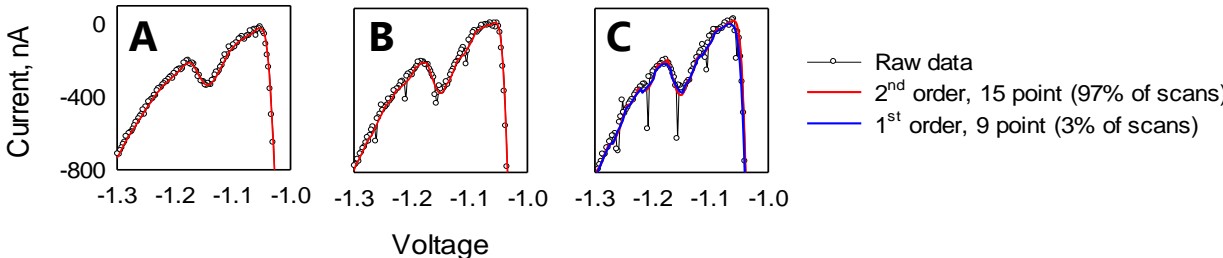



# Figure 3

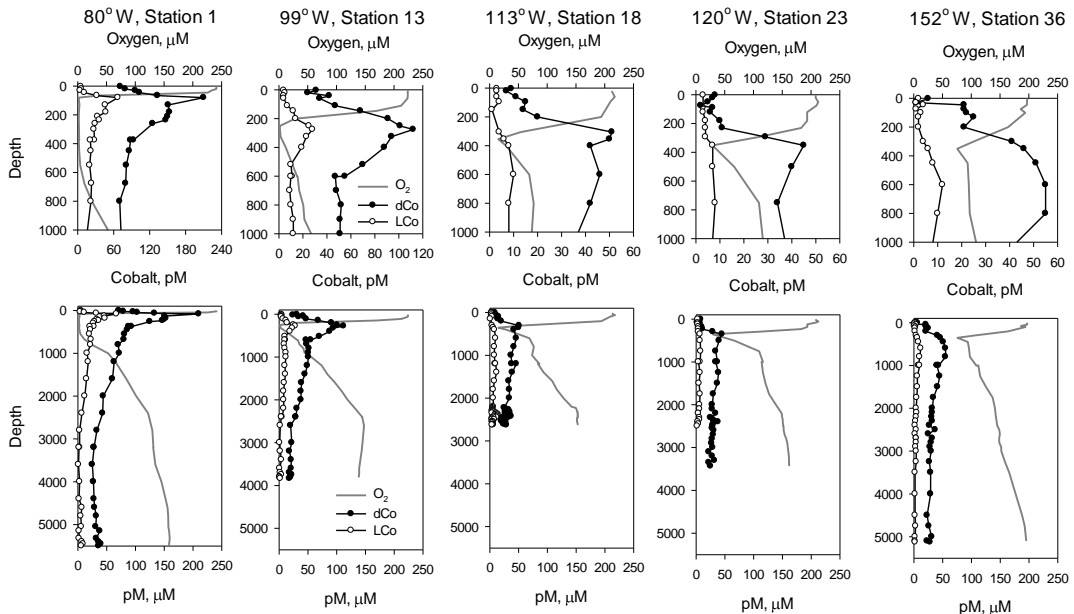





# Figure 4





# Figure 5

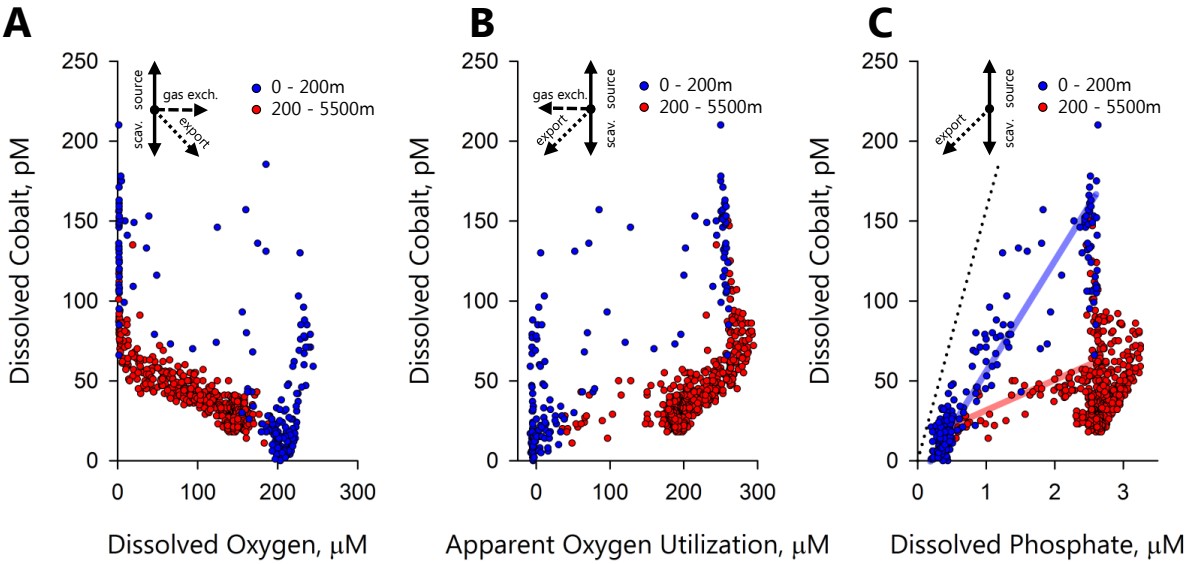



# Figure 6

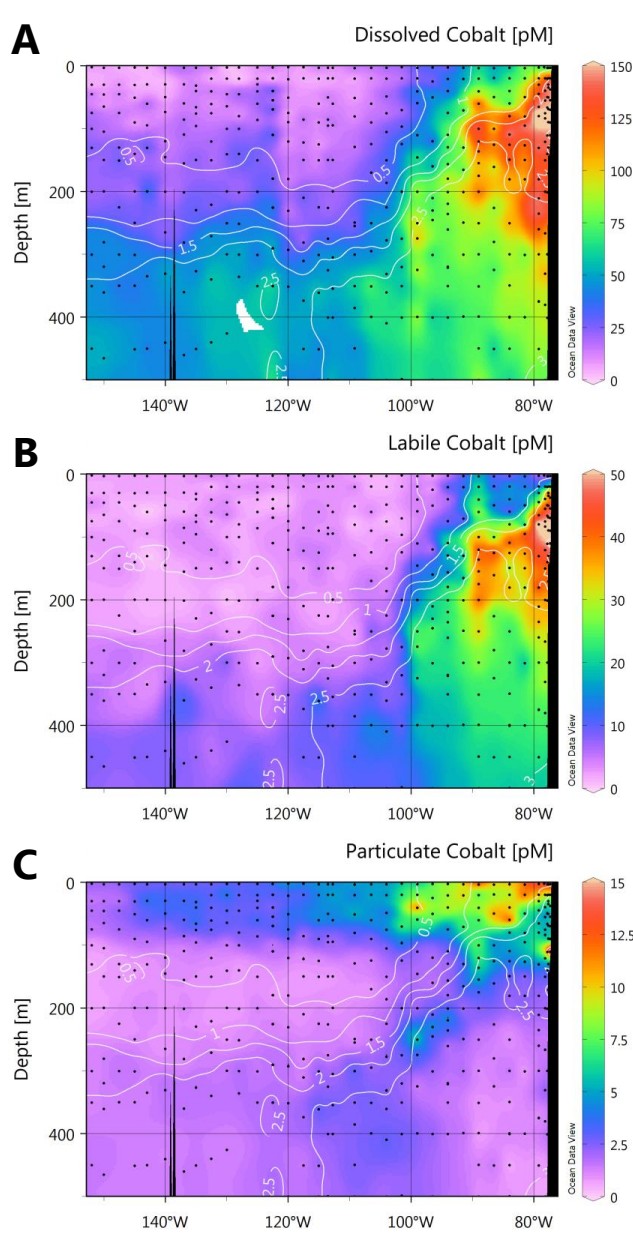





# Figure 7

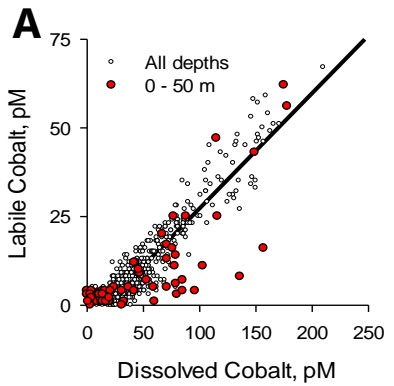
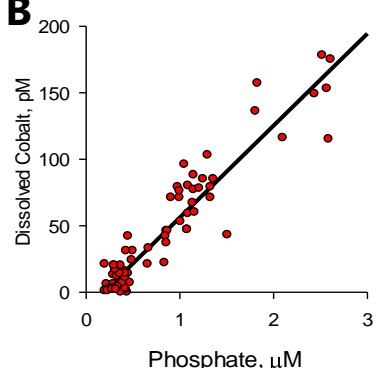
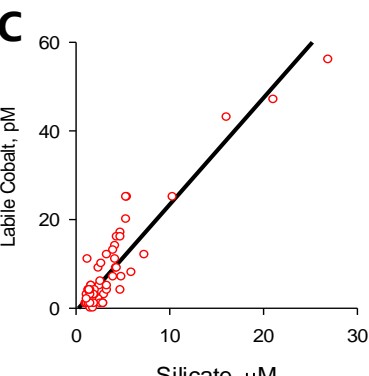



# Figure 8

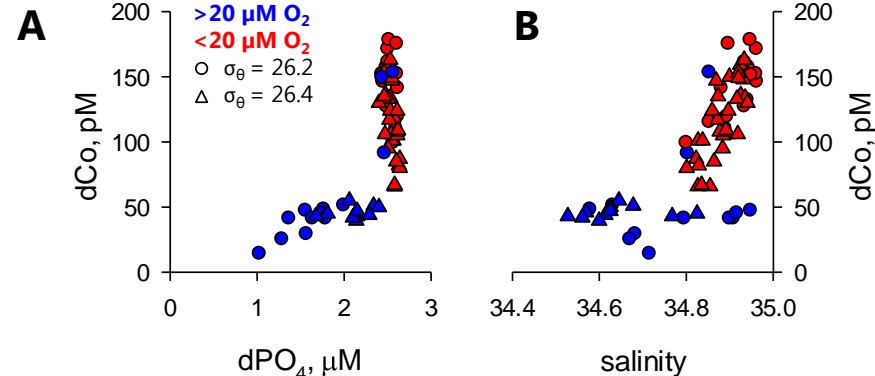



# Figure 9

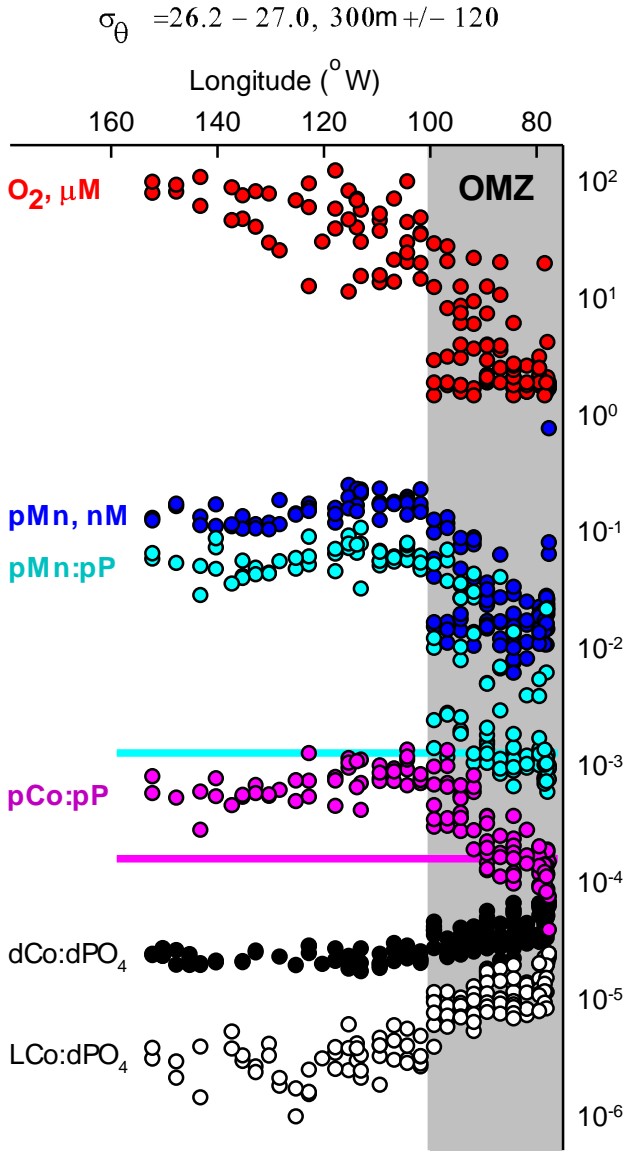



# Figure 10

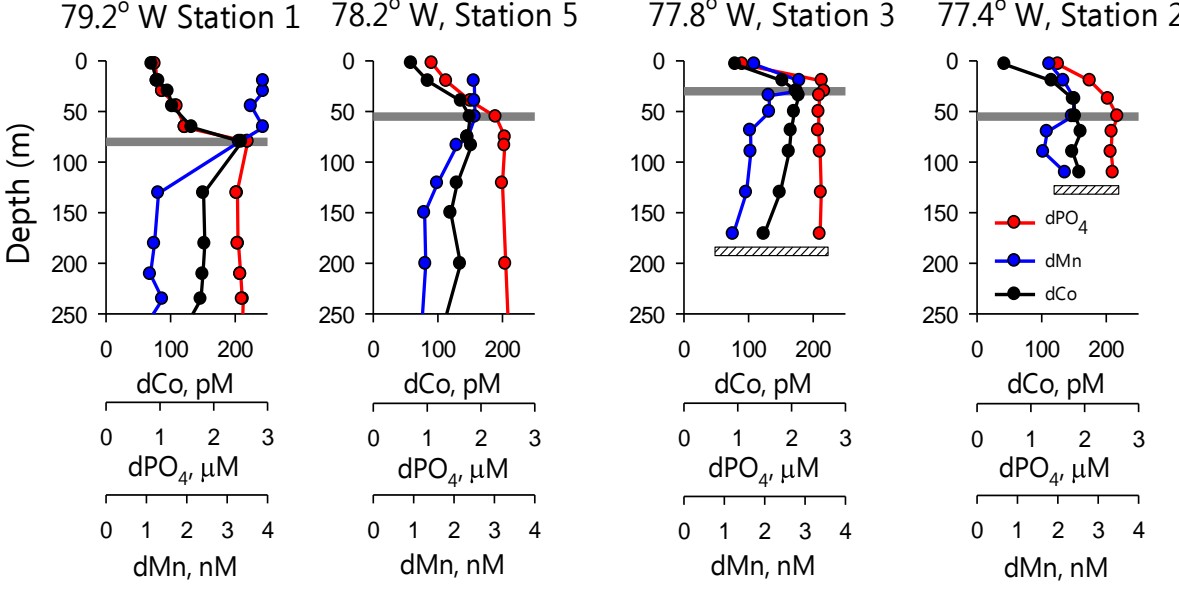




# Figure 11

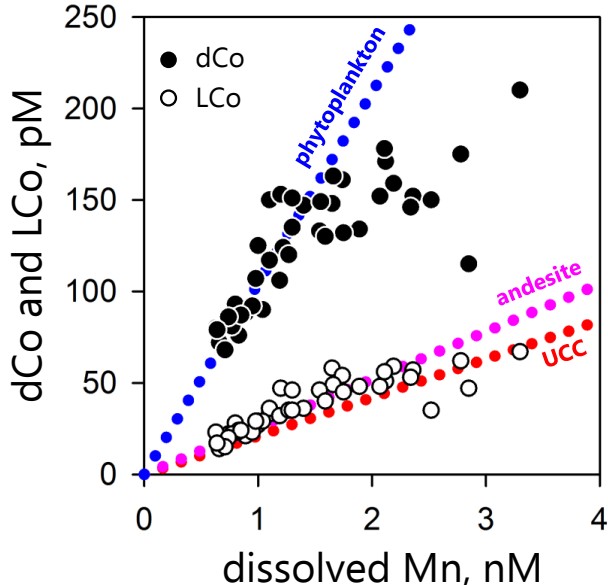



# Figure 12

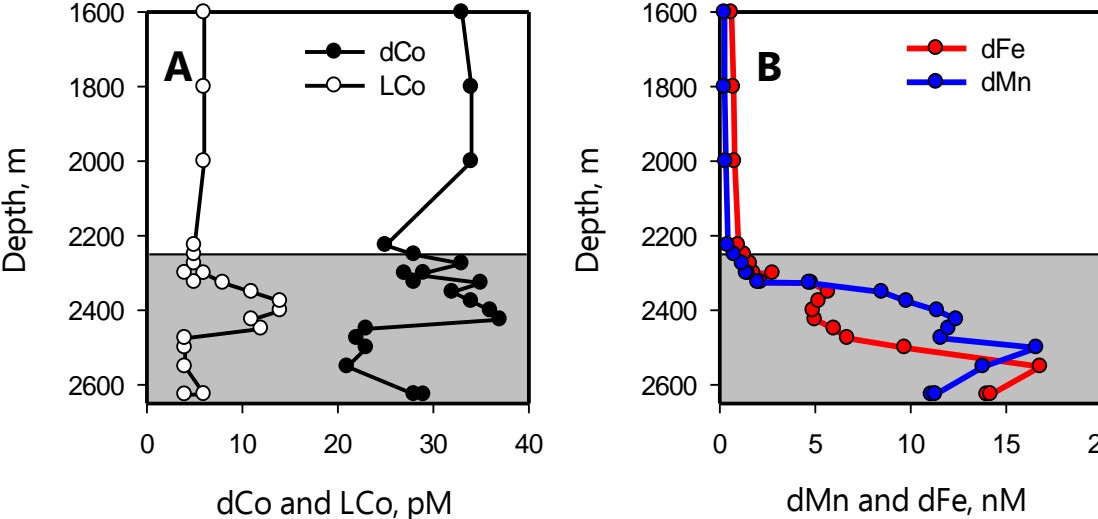



# Figure 13

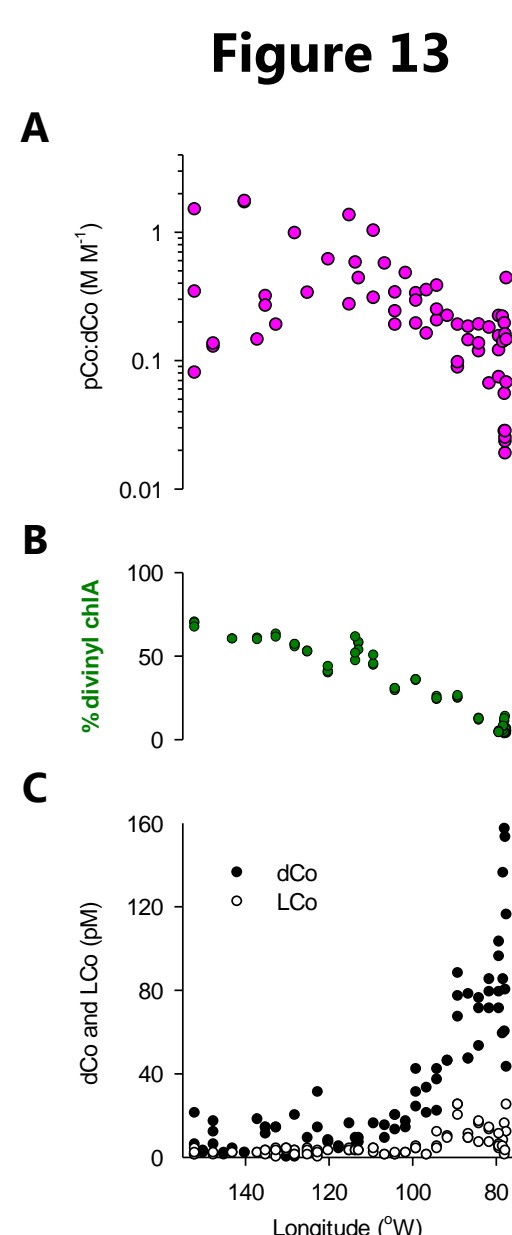




# Figure 14

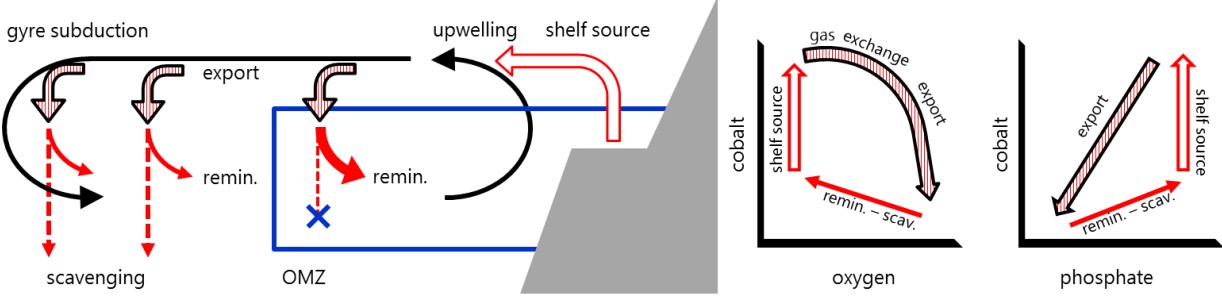