# Peer review of "A dissolved cobalt plume in the oxygen minimum zone of the Eastern Tropical South Pacific N. J. Hawco, 1,2 D.C. Ohnemus, 3 J. A. Resing, 4 B. S. Twining 3 and M. A. Saito 2 1 MIT/WHOI Joint Program in Oceanography/Applied Ocean Science and Engineering, Woods Hole, MA, USA 2"

_Biogeosciences, 2016_

## Referee Comment (RC1) · Anonymous Referee #1 · 7 Jun 2016

General comments "A cobalt plume in the oxygen minimum zone of the Eastern Tropical South Pacific" by N. Hawco et al. is a well written paper of a large dataset from a region with no data published on the same scale. These basin-scale, full depth studies provide invaluable data and insights into the biogeochemical cycling of trace elements in the oceans. My main complaint is that the authors refer forward to figures that haven't been discussed yet, and so I feel the paper needs a bit of restructuring to address this. Other than this I only have a few minor comments/suggestions for the authors' consideration, and recommend this paper for review in Biogeosciences, following minor revisions. Specific comments Line 57. I'm not sure you can definitively say that Co is the least abundant inorganic nutrient, Cd is similarly in the same range, I'd say, "one of the least" Line 64. which suggests... Line 88. 100 pM – the space between the value and unit is missing. This error occurs frequently, but not every time. I have not listed

this observation where it occurs later in the text. Line 170. Include the resistivity of the Milli-Q water here. As Milli-Q is a brand name it might be better to say ultrahigh purity water, or something similar, instead of Milli-Q Line 145. Delete " is measured" Line 203. 1.5 mL of 1.5 M sodium nitrate Line 215. Broader than what? Just "broad" will do, perhaps with the range of tested concentrations stated. Line 216. Replace "deviation" with "variance" Line 234. in the lab Line 253. You should probably include the initials; C. Parker and K. Bruland Line 281. What was the ratio of HCl: NHO3: HF? Fig. 4. I think it would help the discussion to add some station numbers to this figure Line 351. Baars and Croot (2015) Line 410. You shouldn't really be referring forward to Figure 13c here. This needs some rearranging so that you are not referring forward. You could simply use the values without referring to Figure 13c until later in the text. There are a number of instances that you are referring to figures that haven't been described yet, which you should try to avoid as much as possible Line 445. "...new cobalt sourced from the shelf is rapidly incorporated into biological cycling and that the capacity for phytoplankton Co uptake..." - the biological cycle, or biological cycles Line 502. delete "in the" Line 527. Is there any documentation of reducing sediments on the South American continental shelf that could support your assertion? Line 544. Consistent with release Line 547. Is this sentence reversed? "...sedimentation outpaces dissolution of Co and Mn only in very shallow water columns and/or proximal to input, which explains the lack of dissolved benthic maxima for both elements beyond Station 2". If sedimentation outpaced dissolution of Co and Mn in shallow water/close to source, then wouldn't we expect to see no benthic maxima? Line 555. Delete second "should provide" Line 603. As I understood the Noble et al (2012) study, the dCo and LCo plumes were extensive, but the dFe plume was much smaller and the dMn plume wasn't evident, at least in the ODV plot, although they do argue for a sedimentary source for all three elements, explaining the differences in plume areal extent by preferential scavenging of Mn>Fe>Co. This sentence needs rewording to reflect this. Line 619. 20 $\mu$M dissolved O2 Line 627. This is also consistent with Sholkovitz and Copland (1981) who estimated that 97% of Co escapes from freshwater systems (Sholkovitz, E.R., and Copland, D., 1981. The

coagulation, solubility and adsorption properties of Fe, Mn, Cu, Ni, Cd, Co and humic acids in river water. Geochimica et Cosmochimica Acta., 45, 181-189.) Line 683. counterpart? Line 701. Or they can access the Co from the particulate pool? Is there any evidence for this in the literature? Line 706. Prochlorococcus produce ligands too. Might be worth mentioning this as you say that the Prochlorococcus abundance was high Line 729. Delete "of" Line 764: fluctuates References. Check your references as some of them are not displayed properly, e.g. Baxter et al (1998), Line 807, and there are some instances of extra, inconsistent punctuation.

––––––––––––––––––––––––––––

---

## Referee Comment (RC2) · Anonymous Referee #2 · 13 Jul 2016

Hawco and colleagues present a comprehensive and exciting data set for cobalt (Co) and ancillary parameters along the US GEOTRACES transect in the South Pacific Ocean. This manuscript will contribute substantially to our understanding of Co bio-geochemical cycling in the ocean and I am looking forward to seeing it published in Biogeosciences.

I agree with most of the authors' interpretations (see one exception below) and my comments are largely editorial. Due to the comprehensive character of the data set, the manuscript is quite long and in places difficult to follow. Even after my second reading I could not quite figure out the logic behind the order of the different sections. The first section of the discussion discusses basin-wide cobalt-oxygen covariation. The second section deals with cobalt-phosphorus covariation, nutrient uptake and export. Then different redox-sensitive Co sources (oxygen minimum zone sediments; hydrothermal

venting followed by scavenging) are discussed and the final section revisits the role of Co in limiting phytoplankton communities in the surface ocean. I understand that all these aspects and processes are coupled (and therefore difficult to structure) but there may still be a way to bring these things in a more consistent order. Many references to figures are missing or wrong and the overall order of figure appearance does not correspond to the figure numbering (Figure 9 appears before Figure 7 and 8, etc.) which also contributed to my confusion.

I have one important comment regarding the interpretation of sedimentary Co sources within the Peruvian oxygen minimum zone. The authors argue that most of the Co and manganese (Mn) in the oxygen minimum zone water derive from sedimentary sources. However, previous work on Co off the coast of Peru (Böning et al., 2004, GCA; cited in the manuscript) has shown that sediments are depleted in Co and Mn relative to the regional lithogenic background, even at the very sediment surface. Based on this observation, Böning et al. (2004) concluded that most of the Co and Mn delivered from terrigenous sources is not dissolved in the sediment but rather in the water column prior to deposition. The water column Co and Mn profiles in Figure 10 do not provide unequivocal evidence for either sediment or (lower) water column dissolution.

In the case of Mn, previously published data for sediments and pore waters by Scholz et al. (GCA 75, 7257-7276) can be used to differentiate sediment and water column sources. Dissolved benthic Mn fluxes on the Peru shelf range between 10 and 40 nmol/cm2yr. Peru margin sediments have average Mn/Al ratios of 0.0052 (g/kg / g/kg). The average Mn/Al in Andesite (0.0123), the regional lithogenic background, can be used to calculate a MnXS of -0.38 g/kg. That is, 0.38 g Mn is lost per kg of sediment accumulating. Multiplying this MnXS with the sediment mass accumulation rate for shelf sediments off Peru (2.8 x 10ˆ-5 kg/cm2yr) yields a Mn loss flux of -10.6 $\mu$g/cm2yr or -192 nmol/cm2yr. This calculation demonstrates that only 5 to 20 % of the Mn is dissolved in the sediments whereas the remainder must be dissolved in the water column prior to deposition. Given the close coupling of Co and Mn in oxygen minimum

zone sediments (Böning et al., 2004) and waters (see present manuscript) I suggest the authors modify the discussion accordingly.

Minor comments:

Title: The paper contains many more interesting observations (and corresponding interpretations) than just the Co plume in the oxygen minimum zone. Consider a more general title that fully captures the comprehensive character of the presented data set.

Line 47, abstract: Add 'in' before 'oligotrophic regions'.

Line 72: It would be useful to define labile Co and the abbreviation LCo here.

Line 75: Does it matter whether the Mn oxides are formed by bacteria or abiotically? I assume they would be enriched in Co either way (see also Line 497).

Results section: The Results section would be easier to follow if it was divided into subsections, e.g., vertical Co profiles, lateral Co distribution, etc.

Line 293: Add number of gyre station so that it can be easily identified in Figure 3.

Line 299: 'overturning' (typo).

Line 313: Add reference to figure.

Line 321: Upwelling of P-rich and O-poor water 'results' in high dCo? Rephrase for clarity.

Line 329: Figure 13?

Line 340: Add figure reference.

Line 359: "On the GP16 transect . . .".

Line 417: I assume you mean "low O2/high dCo water masses mix with high O2/low dCo water masses". Rephrase for clarity.

Line 424: Is the LCo:O2 trend shown in the figures? If yes, add figure reference; If no,

add 'not shown'.

Line 433: What is meant with "double 0 $\mu$M intercept"? 2 x 0 $\mu$M? Rephrase for clarity.

Line 434: What exactly resembles "profiles of N2 . . ."?

Line 444: The 50 m depth range is not shown with separate symbols in Figure 5c. Later in the text other depth ranges are discussed which are not shown either (Line 457). I suggest adding more color codes to Figure 5c to indicate all the depth ranges and corresponding covariation trends discussed in the text. Otherwise the discussion is difficult to follow.

Line 492: What is meant with 'redox barrier'? Do you invoke a biological process or just that Mn oxides do not form at very low oxygen concentrations?

Line 499: Figure 7C does not show pCo.

Line 502: Remove 'in'.

Line 518: Figure 14 is a summary figure which has not been introduced at this point of the discussion. I recommend referring to actual data here and to restrict references to Figure 14 to the Conclusions section. Section 4.3: See comment above on reductive Mn dissolution in the water column.

Line 545-549: The Peruvian shelf occasionally experiences oxidation events which also favor Mn deposition and burial (see discussion in Scholz et al., 2011).

Line 590: "oxidizing conditions . . . prevent reductive dissolution . . ." is a misleading statement. The sediments at the western Pacific margin are certainly Mn-reducing in the shallow subsurface but the oxic surface sediments prevent diffusive escape of the pore water Co and Mn into the water column.

Line 594-Line 606: Because of low water exchange kinetics, Co is incorporated into pyrite and does not tend to from its own sulfide minerals (Morse & Luther, 1999, GCA 63, 3373-3378).

Line 613: Does 'crust' refer to 'andesite' in this equation. It does not matter what you take but it should be consistent throughout the manuscript.

Conclusions: I really like Figure 14 and therefore recommend to introduce it more explicitly at the beginning of the Conclusions (something like: "the major pattern and underlying processes identified in this contribution are summarized in Figure 14"). The major finding can then be summarized by guiding the reader through Figure 14.

Caption Figure 1: What depth or potential density do the isolines correspond to?

---

## Author Comment (AC1) · 26 Jul 2016

We are grateful for the time and effort extended by both anonymous reviewers, whose close attention to detail in reviewing our manuscript makes their praise all the more meaningful.

As both reviewers identified a need for clarity in structuring the results and discussion section and better ordering of figures, future revisions will focus on a more logical, straightforward presentation of arguments and observations. The revised manuscript will contain a better structured results section, split into three sub-sections for introducing dissolved cobalt, labile dissolved cobalt, and particulate cobalt datasets.

In turn, the discussion section will also be more explicitly ordered, as suggested by Anonymous Reviewer #2. Our intention for the discussion was to justify the importance

of the Peru margin cobalt source by looking broadly at the basin scale cycling of cobalt (and its need for continuous input from low-oxygen regions to maintain steady state), then examine that source more explicitly and compare it to other potential sources, and finally discuss how the intensity of this cobalt source may impact phytoplankton communities. We feel that discussion of the potential for cobalt limitation of phytoplankton growth is more meaningful once the distribution, sinks, and sources of cobalt have been discussed in detail. The revised manuscript will contain a short paragraph at the beginning of the discussion section to explicitly introduce the logic of the discussion section and inform the reader of what topics lie ahead. Current sections will be grouped into three super-sections: 1) oceanographic observations and coupling, 2) Sources of cobalt to the South Pacific, and 3) Cobalt scarcity in the euphotic zone. More generous use of subsections will mark shifts in focus (e.g. coastal Cobalt sources, hydrothermal sources).

Finally, we appreciate Reviewer #2's interest in the nature of the coastal cobalt sources and their informative calculation regarding the pathway of manganese sources in the same region. We agree that sources of Co and Mn appear to be highly coupled in this region, both in our data and in sediment investigations presented by Boning et al. 2004. In a revised draft we would like to make more explicit reference to the Mn porewater fluxes as a constraint on cobalt sources, demonstrated in Reviewer #2's comment. We will clarify our language so that we discuss a "coastal" cobalt source, rather than one explicitly originating from the sediments. However, since sampling on GP16 avoided areas of coastlines directly influenced input of terrigenous sediments, it unresolved whether that the Co and Mn sources do originate from sediments along river deltas or do represent an immediate desorption, as Reviewer #2 implies. Regardless, we appreciate the opportunity to clarify hypotheses about source mechanisms.

Mak Saito and Nick Hawco

Below, we address all explicit line comments from Reviewer 1:

Reviewer 1:

Line 57. I'm not sure you can definitively say that Co is the least abundant inorganic nutrient, Cd is similarly in the same range, I'd say, "one of the least"

Will be replaced. However, while surface Cd concentrations are extremely low, the mean oceanic Co concentration is significantly less than Cd, see Moore et al. 2013.

Line 64. which suggests. . .

Will be changed

Line 88. 100 pM – the space between the value and unit is missing. This error occurs frequently, but not every time. I have not listed this observation where it occurs later in the text.

We apologize for these errors and will comb the manuscript to fix them.

Line 170. Include the resistivity of the Milli-Q water here. As Milli-Q is a brand name it might be better to say ultrahigh purity water, or something similar, instead of Milli-Q

Will change.

Line 145. Delete " is measured"

Will change

Line 203. 1.5 mL of 1.5 M sodium nitrate

Will change. However the reagent is nitrite, not nitrate.

Line 215. Broader than what? Just "broad" will do, perhaps with the range of tested concentrations stated.

Will change

Line 216. Replace "deviation" with "variance"

Will change

Line 234. in the lab

Will change

Line 253. You should probably include the initials; C. Parker and K. Bruland

Will change

Line 281. What was the ratio of HCl: NHO3: HF?

Will add

Fig. 4. I think it would help the discussion to add some station numbers to this figure

We will modify this figure to add station numbers

Line 351. Baars and Croot (2015)

Will correct

Line 410. You shouldn't really be referring forward to Figure 13c here. This needs some rearranging so that you are not referring forward. You could simply use the values without referring to Figure 13c until later in the text. There are a number of instances that you are referring to figures that haven't been described yet, which you should try to avoid as much as possible

We appreciate this suggestion, will remove figure reference and change other figure references accordingly.

Line 445. ". . .new cobalt sourced from the shelf is rapidly incorporated into biological cycling and that the capacity for phytoplankton Co uptake. . ." - the biological cycle, or biological cycles

Line 502. delete "in the"

Will change

Line 527. Is there any documentation of reducing sediments on the South American continental shelf that could support your assertion?

This was address by Reviewer 2 as well. We will add more explicit reference to Scholz et al. 2011 (GCA),which documents metal reducing conditions in porewaters and their diffusive flux out.

Line 544. Consistent with release

Will change.

Line 547. Is this sentence reversed? ". . .sedimentation outpaces dissolution of Co and Mn only in very shallow water columns and/or proximal to input, which explains the lack of dissolved benthic maxima for both elements beyond Station 2". If sedimentation outpaced dissolution of Co and Mn in shallow water/close to source, then wouldn't we expect to see no benthic maxima?

Will change to "Most Co and Mn appears to be released directly to the water column rather than into sediment pore waters, although diffusive fluxes from porewaters are significant in very shallow water columns (<150m, Scholz et al. 2011, Boning et al. 2004).

Line 555. Delete second "should provide"

Will change

Line 603. As I understood the Noble et al (2012) study, the dCo and LCo plumes were extensive, but the dFe plume was much smaller and the dMn plume wasn't evident, at least in the ODV plot, although they do argue for a sedimentary source for all three elements, explaining the differences in plume areal extent by preferential scavenging of Mn>Fe>Co. This sentence needs rewording to reflect this.

This is correct that the dFe plume was much smaller than the dCo plume. We will clarify that it is smaller, in part due to faster scavenging kinetics for dFe. For Mn the

plume overlaps with the large surface maxima making the OMZ plume more difficult to observe. Moreover the larger abundances of Mn and its presumably labile form make it susceptible to faster removal than Co (which is scarcer and can be complexed).

Line 619. 20 $\mu$M dissolved O2

Will change

Line 627. This is also consistent with Sholkovitz and Copland (1981) who estimated that 97% of Co escapes from freshwater systems (Sholkovitz, E.R., and Copland, D., 1981. The coagulation, solubility and adsorption properties of Fe, Mn, Cu, Ni, Cd, Co and humic acids in river water. Geochimica et Cosmochimica Acta., 45, 181-189.)

We will include this reference. Reviewer #2 also suggested adding more nuance to our discussion of coastal cobalt source and the revised paper will mention the lack of Co coagulation in estuaries reported by Sholkovitz and Copland.

Line 683. counterpart?

Will change

Line 701. Or they can access the Co from the particulate pool? Is there any evidence for this in the literature?

This is an interesting point. As the particulate Co pool seems to be dominated by biomass, it is likely that much of the particulate dCo is continually recycled, as with other nutrients. We will add this to our list of possibilities outlined in this paragraph .It is somewhat uncertain, however, if the particulate biomass dCo represents free Co metal (i.e. labile Co) or strongly bound Co (such as cobalamins).

Line 706. Prochlorococcus produce ligands too. Might be worth mentioning this as you say that the Prochlorococcus abundance was high

We will add mention to this.

Line 729. Delete "of"

Will change.

Line 764: fluctuates

Will change.

References. Check your references as some of them are not displayed properly, e.g. Baxter et al (1998), Line 807, and there are some instances of extra, inconsistent punctuation.

We appreciate the detail undertaken in this review and will remove existing errors in reference formatting and punctuation.

---

## Author Comment (AC2) · 26 Jul 2016

See combined response to reviewers in response to reviewer #1.

Specific responses to Reviewer #2

Title: The paper contains many more interesting observations (and corresponding interpretations) than just the Co plume in the oxygen minimum zone. Consider a more general title that fully captures the comprehensive character of the presented data set.

This is a valid point, and there are advantages to having papers with long complex titles versus shorter memorable ones. We will consider this point further.

Line 47, abstract: Add 'in' before 'oligotrophic regions'.

Will change

Line 72: It would be useful to define labile Co and the abbreviation LCo here.

We appreciate the suggestion. We will put the definition for labile Cobalt here.

Line 75: Does it matter whether the Mn oxides are formed by bacteria or abiotically? I assume they would be enriched in Co either way (see also Line 497).

Perhaps not in the scope of this manuscript, but biogenesis or abiogenesis of Mn oxides have important implications for the sensitivity of Co and Mn scavenging in the past and future.

Results section: The Results section would be easier to follow if it was divided into subsections, e.g., vertical Co profiles, lateral Co distribution, etc.

We plan to organize the results section into more subsections.

Line 293: Add number of gyre station so that it can be easily identified in Figure 3.

Will change.

Line 299: 'overturning' (typo).

Will change.

Line 313: Add reference to figure.

Will change.

Line 321: Upwelling of P-rich and O-poor water 'results' in high dCo? Rephrase for clarity.

Will change.

Line 329: Figure 13?

Will remove reference to this figure.

Line 340: Add figure reference.

Will add.

Line 359: "On the GP16 transect . . .".

Will add.

Line 417: I assume you mean "low O2/high dCo water masses mix with high O2/low dCo water masses". Rephrase for clarity.

This is correct, will rephrase.

Line 424: Is the LCo:O2 trend shown in the figures? If yes, add figure reference; If no, add 'not shown'.

Will add not shown.

Line 433: What is meant with "double 0 $\mu$M intercept"? 2 x 0 $\mu$M? Rephrase for clarity.

Will rephrase. "Double" in this case referred to the actual dCo concentrations at 0 umol O2 being more than twice the concentration of what the intercept of the linear Co:O2 regression predicted.

Line 434: What exactly resembles "profiles of N2 . . ."?

Will add clause describing how the profiles are similar. What is similar about them is the fact that both dCo and excess N2 from denitrification both peak just below the oxycline rather than having a consistent value throughout the entire depth range of anoxia. Because excess N2 results, in part, from heterotrophic remineralization within the upper depth strata of the OMZ, the similarity between N2 profiles and dCo in this region imply that the dCo maximum reflects remineralization of sinking biomass.

Line 444: The 50 m depth range is not shown with separate symbols in Figure 5c. Later in the text other depth ranges are discussed which are not shown either (Line 457). I suggest adding more color codes to Figure 5c to indicate all the depth ranges and corresponding covariation trends discussed in the text. Otherwise the discussion

is difficult to follow.

This is a good suggestion and we will modify the figures in accordance with the three depth ranges discussed in the text.

Line 492: What is meant with 'redox barrier'? Do you invoke a biological process or just that Mn oxides do not form at very low oxygen concentrations?

The instability of Mn oxides at low oxygen is the simplest explanation. We will rephrase to indicate that the particulate dataset is consistent with thermodynamic and kinetic arguments against Mn-oxidation in OMZs.

Line 499: Figure 7C does not show pCo.

This is correct. Will change to figure 6C.

Line 502: Remove 'in'.

Will change.

Line 518: Figure 14 is a summary figure which has not been introduced at this point of the discussion. I recommend referring to actual data here and to restrict references to Figure 14 to the Conclusions section.

This is an excellent recommendation, we will reserve figure 14 for the conclusion section.

Section 4.3: See comment above on reductive Mn dissolution in the water column.

Yes, we plan to clarify that a coastal Mn and Co source does not appear to originate in the sediments underlying the stations reported here, leveraging the calculations reviewer #2 introduced from the Scholz et al. 2011 study.

Line 545-549: The Peruvian shelf occasionally experiences oxidation events which also favor Mn deposition and burial (see discussion in Scholz et al., 2011).

This is an important point that will be added to this section. This may very well by why

only the shallowest station has a benthic Co and Mn maximum!

Line 590: "oxidizing conditions . . . prevent reductive dissolution . . ." is a misleading statement. The sediments at the western Pacific margin are certainly Mn-reducing in the shallow subsurface but the oxic surface sediments prevent diffusive escape of the pore water Co and Mn into the water column.

We appreciate this point and will rewrite to differentiate between in-situ redox conditions (downcore) in the sediment and the mass transfer of Co and Mn from such sediments to the open ocean, which appear to be limited due to re-precipitation in surface sediments and in the water column.

Line 594-Line 606: Because of low water exchange kinetics, Co is incorporated into pyrite and does not tend to from its own sulfide minerals (Morse & Luther, 1999, GCA 63, 3373-3378).

We are thankful for referral to this reference. We will modify this paragraph to indicate incorporation into pyrite rather than direct CoS precipitation.

Line 613: Does 'crust' refer to 'andesite' in this equation. It does not matter what you take but it should be consistent throughout the manuscript.

The range in calculated fluxes represents calculations from both upper continental crust, as defined by McLennan 2001, and Andesitic endmembers, as defined by Taylor and McLennan 1995).We will make this more explicit in the paragraph.

Conclusions: I really like Figure 14 and therefore recommend to introduce it more explicitly at the beginning of the Conclusions (something like: "the major pattern and underlying processes identified in this contribution are summarized in Figure 14"). The major finding can then be summarized by guiding the reader through Figure 14.

We appreciate this suggestion and hold reference to figure 14 until the conclusions, where we will explicitly introduce it as a conceptual model for the Co cycle here.

Caption Figure 1: What depth or potential density do the isolines correspond to?

This represents oxygen at 300m, which was accidentally omitted from the caption. We will remedy this.
* * *

---

## Author Response (AR1)

To the editor,

We are grateful for the time and effort extended by both anonymous reviewers, whose close attention to detail in reviewing our manuscript makes their praise all the more meaningful.

Both reviewers identified a need for clarity in structuring the results and discussion section and better ordering of figures (and where/when they are referenced in the discussion). To remedy this, have split our results section into 3 subsections, and our Discussion section now contains 10 subsections. We have also added an introductory paragraph at the start of the discussion that outlines the upcoming discussion in a logical order. We hope that this new ordering of the manuscript will assist readers in approaching this large manuscript. A great deal of care has also been taken to make sure figures are not referenced out of order, which was an annoyance to both reviewers.

Furthermore, the discussion of coastal cobalt sources has been modified to reflect an ambiguity in the actual source mechanism outlined in Reviewer 2's comments. We have added discussion, particularly in lines 620-626 and 699-708, which acknowledges the lack of mass balance between Mn/Al accumulation rates off Peru and diffusive Mn fluxes out of the sediments. We have also clarified our language to present this source as "coastal" rather than "sedimentary", so that the ambiguity of the Mn and Co sources is preserved.

We have also reorganized the conclusion section, as recommended by reviewer 2, so that figure 14 can be used as a guide to the summary of the data presented in the Results and Discussion sections.

Below, we describe changes in the new manuscript from both Reviewers' line comments. The line in the full-markup draft is present in parentheses. Additional changes to the manuscript are largely to make sentences more easily understood, and, with the exception of the description of the coastal source section, none of the conclusions have been altered.

Sincerely,

Nick Hawco and Mak Saito

**Reviewer #1**

**Line 57. I'm not sure you can definitively say that Co is the least abundant inorganic nutrient, Cd is similarly in the same range, I'd say, "one of the least"**

(59) This has been replaced. However, while surface Cd concentrations are extremely low, the mean oceanic Co concentration is significantly less than Cd, see Moore et al. 2013.

**Line 64. which suggests. . .**

(66) This has been changed.

**Line 88. 100 pM – the space between the value and unit is missing. This error occurs frequently, but not every time. I have not listed this observation where it occurs later in the text.**

(91) This has been changed, and we have made our best attempt to correct similar mistakes throughout the manuscript.

**Line 170. Include the resistivity of the Milli-Q water here. As Milli-Q is a brand name it might be better to say ultrahigh purity water, or something similar, instead of Milli-Q**

(174) This has been changed, and other references to Milli-Q water have been substituted with 18 MΩ water. See lines 179, 212, 213.

**Line 145. Delete " is measured"**

(149) This has been deleted.

**Line 203. 1.5 mL of 1.5 M sodium nitrate**

(207) This has been changed to better separate the volume and the concentration of reagents. However the reagent is nitrite, not nitrate.

**Line 215. Broader than what? Just "broad" will do, perhaps with the range of tested concentrations stated.**

(219) Broader has been changed to broad.

**Line 216. Replace "deviation" with "variance"**

(220) This has been changed.

**Line 234. in the lab**

(239) "the" has been added to this line.

**Line 253. You should probably include the initials: C. Parker and K. Bruland**

(259) We have added initials.

**Line 281. What was the ratio of HCl: NHO3: HF?**

(285) The concentrations of the individual acids have been added. For reference the ratio is 1:1:1, which is clearer in the new version.

**Fig. 4. I think it would help the discussion to add some station numbers to this figure**

(Fig. 4) We have added station numbers that appear in figure 1 to the top of the sections in Figure 4. We have done the same for figure 6.

**Line 351. Baars and Croot (2015)**

(365) This mistake has been corrected.

**Line 410. You shouldn't really be referring forward to Figure 13c here. This needs some rearranging so that you are not referring forward. You could simply use the values without referring to Figure 13c until later in the text. There are a number of instances that you are**

**referring to figures that haven't been described yet, which you should try to avoid as much as possible**

(341) We have removed the reference to this figure here. For this version of the manuscript, we have worked hard to avoid the need to reference figures out of order and have reworked the layout of both the results and the discussion section to achieve this.

**Line 445. ". . .new cobalt sourced from the shelf is rapidly incorporated into biological cycling and that the capacity for phytoplankton Co uptake. . ." - the biological cycle, or biological cycles**

(485) this phrase has been changed to "the biological cycle"

**Line 502. delete "in the"**

(546) This error has been fixed.

**Line 527. Is there any documentation of reducing sediments on the South American continental shelf that could support your assertion?**

(573) We have removed reference to the reducing sediments in this sentence. Due to similar comments from Reviewer 2, we have added an additional paragraph discussing source mechanisms in detail (Lines 611–626) that makes better use of the existing literature in documenting reducing conditions on the Peru shelf, including the Scholz et al. 2011 GCA study.

**Line 544. Consistent with release**

(618) "with" has been added

**Line 547. Is this sentence reversed? ". . .sedimentation outpaces dissolution of Co and Mn only in very shallow water columns and/or proximal to input, which explains the lack of dissolved benthic maxima for both elements beyond Station 2". If sedimentation outpaced dissolution of Co and Mn in shallow water/close to source, then wouldn't we expect to see no benthic maxima?**

(611-626) To avoid unnecessary confusion to the reader, we have deleted this sentence. Reorganization of this section has largely been to acknowledge that, while a margin cobalt source is certain, its mechanism is unclear. In addition to rewriting this paragraph, we have also added a new paragraph (Lines 698-707) that highlights this uncertainty more explicitly.

**Line 603. As I understood the Noble et al (2012) study, the dCo and LCo plumes were extensive, but the dFe plume was much smaller and the dMn plume wasn't evident, at least in the ODV plot, although they do argue for a sedimentary source for all three elements, explaining the differences in plume areal extent by preferential scavenging of Mn>Fe>Co. This sentence needs rewording to reflect this.**

(694) We have rewritten this sentence so that it is solely focused on the Co plume. For the scope of this section, the main importance is that both the South Pacific and South Atlantic have OMZ plumes of dCo and both regions have depleted Co contents in shelf sediments, suggesting that the same mechanisms is acting in both basins.

**Line 619. 20 μM dissolved O2**

(722) This has been changed.

**Line 627. This is also consistent with Sholkovitz and Copland (1981) who estimated that 97% of Co escapes from freshwater systems (Sholkovitz, E.R., and Copland, D., 1981. The coagulation, solubility and adsorption properties of Fe, Mn, Cu, Ni, Cd, Co and humic acids in river water. Geochimica et Cosmochimica Acta., 45, 181-189.)**

(96) We have added this reference to the introductory paragraph describing coastal Co sources (Line 96). While this is a very relevant study, in light of the uncertainty in the actual source mechanism we found it more appropriate in the introduction than in the discussion.

**Line 683. counterpart?**

This has been changed.

**Line 701. Or they can access the Co from the particulate pool? Is there any evidence for this in the literature?**

(811-813) We have modified this sentence to explicitly refer to the particulate pool. Due to the low abundance of Co in dust and the extremely low dust supply to the south pacific, we interpret the particulate Co in the surface ocean (Fig 6c) to reflect Co bound in biomass. We therefore interpret the high pCo:dCo ratio in the South Pacific gyre to reflect recycling between Co-bearing biomolecules as the plankton biomass turns over, rather than the existence of an unexploited, additional resource (such as dust).

**Line 706. Prochlorococcus produce ligands too. Might be worth mentioning this as you say that the Prochlorococcus abundance was high**

(820) We added a mention to this observation.

**Line 729. Delete "of"**

(843-44) This has been changed to "in the Peru upwelling region"

**Line 764: fluctuates**

(893) This has been changed.

**References. Check your references as some of them are not displayed properly, e.g. Baxter et al (1998), Line 807, and there are some instances of extra, inconsistent punctuation.**

(938) We have corrected the Baxter et al. reference and have tried to correct similar errors in formatting.

**Specific responses to Reviewer #2**

**Title: The paper contains many more interesting observations (and corresponding interpretations) than just the Co plume in the oxygen minimum zone. Consider a more general title that fully captures the comprehensive character of the presented data set.**

After much thought, we have decided to keep the shorter, succinct title of the manuscript. Almost all of the observations are focused around the oxygen minimum zone and the processes supporting the cobalt plume therein. We have tried to emphasize the role of the OMZ plume in Cobalt biogeochemistry in the first paragraph of the discussion section, which have been added since review (Lines 396-419)

**Line 47, abstract: Add 'in' before 'oligotrophic regions'.**

(47) This has been added.

**Line 72: It would be useful to define labile Co and the abbreviation LCo here.**

(75) We have moved up the definition of LCo as suggested.

**Line 75: Does it matter whether the Mn oxides are formed by bacteria or abiotically? I assume they would be enriched in Co either way (see also Line 497).**

(78) We have changed 'bacterial' to 'authigenic.'

**Results section: The Results section would be easier to follow if it was divided into subsections, e.g., vertical Co profiles, lateral Co distribution, etc.**

We have split the results section into 3 major subsections discussing the distribution of dCo (300), pCo (347), and LCo (362).

**Line 293: Add number of gyre station so that it can be easily identified in Figure 3.**

(305) Stations plotted in figure 3 have been added in parentheses. Per a comment from Reviewer 1, we have also added station numbers to figures 4 and 6.

**Line 299: 'overturning' (typo).**

(310) Corrected.

**Line 313: Add reference to figure.**

In order to avoid for referencing figure 10 out of order (an issue raised by both reviewers), we have not made this change. However, we have marked Station 2 on both figures 4 and 6 (and figure 1) to make it easier for a reader to understand.

**Line 321: Upwelling of P-rich and O-poor water 'results' in high dCo? Rephrase for clarity.**

(333-335) We have rewritten this sentence for increased clarity.

**Line 329: Figure 13?**

(342) We have removed reference to this figure.

**Line 340: Add figure reference.**

Will add.

**Line 359: "On the GP16 transect . . .".**

(353) This has been added.

**Line 417: I assume you mean "low O2/high dCo water masses mix with high O2/low dCo water masses". Rephrase for clarity.**

(452-453)This sentence has been rewritten to make its meaning more clear.

**Line 424: Is the LCo:O2 trend shown in the figures? If yes, add figure reference; If no, C3 add 'not shown'.**

(460) 'not shown' was added.

**Line 433: What is meant with "double 0 μM intercept"? 2 x 0 μM? Rephrase for clarity.**

(469) This has been rephrased. The word "twice" has been substituted for "double" in the original manuscript, which was the main cause of confusion.

**Line 434: What exactly resembles "profiles of N2 . . ."?**

(471-476) The wording of this sentence has been redone. It has been emphasized that the dCo profile is what is similar to the excess N2 profile.

**Line 444: The 50 m depth range is not shown with separate symbols in Figure 5c. Later in the text other depth ranges are discussed which are not shown either (Line 457). I suggest adding more color codes to Figure 5c to indicate all the depth ranges and corresponding covariation trends discussed in the text. Otherwise the discussion is difficult to follow.**

(485) We have modified figure 5 so that the 0-50m depth range is presented as cyan, and the 50-200m depth range is presented as blue. Specific references to the colors in figure 5 and also figure 9 have been added parenthetically (also Lines 498, 499, 503, 535, 541, 547, 554, 556…).

**Line 492: What is meant with 'redox barrier'? Do you invoke a biological process or just that Mn oxides do not form at very low oxygen concentrations?**

(536) We have rewritten so be more specific, mainly highlighting the thermodynamic and kinetic basis for a sensitivity of Mn oxidation to changes in O2 concentration.

**Line 499: Figure 7C does not show pCo.**

(545) We have changed to figure reference to 6C.

**Line 502: Remove 'in'.**

(547) This has been changed.

**Line 518: Figure 14 is a summary figure which has not been introduced at this point of the discussion. I recommend referring to actual data here and to restrict references to Figure 14 to the Conclusions section.**

(563) We have removed the reference to figure 14, both here and everywhere else, with the exception of the conclusion section, which has been reorganized. For the purposes of this sentence, we have added a reference to figure 5c.

**Section 4.3: See comment above on reductive Mn dissolution in the water column.**

(620-627) We have incorporated the conclusions from the Scholz et al. 2011 study more explicitly in this section, and in the next (Lines 699-708). We have backed off of the wording on sedimentary sources, instead using "coastal" as a kind of catch-all until more details about the nature of this source come to light.

**Line 545-549: The Peruvian shelf occasionally experiences oxidation events which also favor Mn deposition and burial (see discussion in Scholz et al., 2011).**

(682-684) We have added a sentence making explicit mention to this effect.

**Line 590: "oxidizing conditions . . . prevent reductive dissolution . . ." is a misleading statement. The sediments at the western Pacific margin are certainly Mn-reducing in the shallow subsurface but the oxic surface sediments prevent diffusive escape of the pore water Co and Mn into the water column.**

(677-678) We have clarified our hypothesis here, emphasizing that oxidizing conditions limit the release of Co from the sediments, as point out in the above comment.

**Line 594-Line 606: Because of low water exchange kinetics, Co is incorporated into pyrite and does not tend to from its own sulfide minerals (Morse & Luther, 1999, GCA 63, 3373-3378).**

(685) We have have made note of this distinction and have modified our language accordingly. The Morse and Luther citation is added to Lines 688-689.

**Line 613: Does 'crust' refer to 'andesite' in this equation. It does not matter what you take but it should be consistent throughout the manuscript.**

(722) The range in calculated fluxes represents calculations from both upper continental crust, as defined by McLennan 2001, and Andesitic endmembers, as defined by Taylor and McLennan 1995).We have tried to make this more explicit in the paragraph by indicating this in parenthesis, as well as an additional reference to table 2.

**Conclusions: I really like Figure 14 and therefore recommend to introduce it more explicitly at the beginning of the Conclusions (something like: "the major pattern and underlying processes identified in this contribution are summarized in Figure 14"). The major finding can then be summarized by guiding the reader through Figure 14.**

(853) We have reorganized the conclusion section, as suggested, by way of examining figure 14. The ordering of the summary follows an approximate 'source-to-sink' direction which can be followed along in figure 14.

**Caption Figure 1: What depth or potential density do the isolines correspond to? This represents oxygen at 300m, which was accidentally omitted from the caption.**

(1205) We have added that the O2 data comes from 300m.

[revised manuscript text omitted]